# Impacts of Different Characterizations of Large-Scale Background on Simulated Regional-Scale Ozone Over the Continental United States

Christian Hogrefe[1], Peng Liu[2], George Pouliot[1], Rohit Mathur[1], Shawn Roselle[1], Johannes Flemming[3], Meiyun Lin[4,5], and Rokjin J. Park[6]

[1]Computational Exposure Division, National Exposure Research Laboratory, U.S. Environmental Protection Agency, RTP, NC, USA
[2]National Research Council Fellow at National Exposure Research Laboratory, U.S. Environmental Protection Agency, RTP, NC, USA
[3]European Centre for Medium-Range Weather Forecasts, Reading, U.K.
[4]Atmospheric and Oceanic Sciences, Princeton University, Princeton, NJ 08540, USA
[5]NOAA Geophysical Fluid Dynamics Laboratory, Princeton, NJ 08540, USA
[6]School of Earth and Environmental Sciences, Seoul National University, Seoul, Republic of Korea

*Correspondence to*: Christian Hogrefe (hogrefe.christian@epa.gov)

**Abstract.** This study analyzes simulated regional-scale ozone burdens both near the surface and aloft, estimates process contributions to these burdens, and calculates the sensitivity of the simulated regional-scale ozone burden to several key model inputs with a particular emphasis on boundary conditions derived from hemispheric or global scale models. The Community Multiscale Air Quality (CMAQ) model simulations supporting this analysis were performed over the continental U.S. for the year 2010 within the context of the Air Quality Model Evaluation International Initiative (AQMEII) and Task Force on Hemispheric Transport of Air Pollution (TF-HTAP) activities. CMAQ Process Analysis (PA) results highlight the dominant role of horizontal and vertical advection on the ozone burden in the mid-to-upper troposphere and lower stratosphere. Vertical mixing, including mixing by convective clouds, couple fluctuations in free tropospheric ozone to ozone in lower layers. Hypothetical bounding scenarios were performed to quantify the effects of emissions, boundary conditions, and ozone dry deposition on the simulated ozone burden. Analysis of these simulations confirms that the characterization of ozone outside the regional-scale modeling domain can have a profound impact on simulated regional-scale ozone. This was further investigated by using data from four hemispheric or global modeling systems (Chemistry – Integrated Forecasting Model (C-IFS), CMAQ extended for hemispheric applications (H-CMAQ), GEOS-Chem, and AM3) to derive alternate boundary conditions for the regional-scale CMAQ simulations. The regional-scale CMAQ simulations using these four different boundary conditions showed that the largest ozone abundance in the upper layers was simulated when using boundary conditions from GEOS-Chem, followed by the simulations using C-IFS, AM3, and H-CMAQ boundary conditions, consistent with the analysis of the ozone fields from the global models along the CMAQ boundaries. Using boundary conditions from AM3 yielded higher springtime ozone columns burdens in the mid- and lower troposphere compared to boundary conditions from the other models. For surface ozone, the differences between the AM3-driven CMAQ simulations and the CMAQ simulations driven by other large-scale models are especially pronounced during spring and winter where they can reach more

than 10 ppb for seasonal mean ozone mixing ratios and as much as 15 ppb for domain-averaged daily maximum 8-hr average ozone on individual days. In contrast, the differences between the C-IFS, GEOS-Chem, and H-CMAQ driven regional-scale CMAQ simulations are typically smaller. Comparing simulated surface ozone mixing ratios to observations and computing seasonal and regional model performance statistics revealed that boundary conditions can have a substantial impact on model performance. Further analysis showed that boundary conditions can affect model performance across the entire range of the observed distribution, although the impacts tend to be lower during summer and for the very highest observed percentiles. The results are discussed in the context of future model development and analysis opportunities.

## 1 Introduction

Regional-scale air quality modeling systems such as the Community Multiscale Air Quality (CMAQ) model (Byun and Schere, 2006), the Comprehensive Air Quality Model with Extensions (CAMx) (Environ, 2014), the Weather Research and Forecasting model coupled to Chemistry (WRF-Chem) (Chapman et al., 2009), and CHIMERE (Vautard et al., 2001) are routinely used for air quality forecasting and planning applications. Many of these models trace their heritage to local-scale models developed to better understand and mitigate elevated ozone in highly polluted urban airsheds such as the Los Angeles basin (McRae and Seinfeld, 1983; Harley et al., 1993). As further research highlighted regional aspects of ozone pollution such as multi-state transport of ozone and its precursors (Eder et al., 1994; Vukovich, 1995; Schichtel and Husar, 2001), these urban-scale models were expanded to represent processes relevant to regional- and continental scale air quality. Because of their origin in urban- and regional-scale air quality modeling and their primary application focus of simulating air quality as it relates to human health (i.e. air applications for air quality planning and forecasting), the performance of these modeling systems is often evaluated primarily at the surface against measurements from monitors in the vicinity of populated areas (Simon et al., 2013; Appel et al, 2017).

The evaluation and intercomparison of regional-scale air quality models has been the central focus of the Air Quality Evaluation International Initiative (AQMEII) that was initiated in 2009 (Rao et al., 2011). Much of the initial work under AQMEII focused on operational model evaluation (Solazzo et al., 2012a,b; Im et al., 2015a,b) while there was an increasing emphasis on diagnostic evaluation in more recent analyses (Solazzo and Galmarini, 2016; Solazzo et al., 2017a,b). Some of these diagnostic analyses have pointed to external model inputs, in particular emissions and boundary conditions representing the larger-scale atmospheric background, as key sources of model error (Schere et al., 2012; Giordano et al., 2015; Solazzo et al., 2017a,b).

Somewhat in parallel to the increased development and use of regional-scale air quality models for air quality management and forecasting starting in the mid-to-late 1990s and early 2000s, there also was active development of global-scale chemistry-transport models such as GEOS-Chem (Bey et al., 2001), the Model for Ozone and Related chemical Tracers (MOZART)

(Horowitz et al., 2003; Emmons et al., 2010), and AM3 (Donner et al., 2011; Lin et al., 2012a) as well as on-line coupled weather-chemistry models such as the European Center for Medium Range Weather Forecasts (ECMWF) Composition – Integrated Forecast System (C-IFS) model (Flemming et al., 2015). A primary use of such global models has been to better understand long-term trends and variability in tropospheric pollutant burdens and budgets and to quantify intercontinental transport. Such research on intercontinental transport of air pollution (Jacob et al., 1999; Li et al., 2002; Holloway et al., 2003; Fiore et al., 2009; Reidmiller et al., 2009; Lin et al., 2015; 2017) led to the increasing recognition of surface ozone as a pollutant that is impacted by phenomena occurring on spatial scales ranging from local to global and temporal scales ranging from hours to decades. Much of this research either contributed to or was directly organized through the Task Force on Hemispheric Transport of Air Pollution (TF-HTAP), resulting in a comprehensive assessment of the science underlying long-range pollutant transport (TF-HTAP, 2010). Model evaluation performed for such global models often has focused on remote, rural, and/or high elevation sites since the grid resolution employed in these models is not expected to fully resolve more fine-scale physical and chemical processes that are important in areas of complex terrain, land/sea interfaces or areas of large emission gradients.

The growing realization that regional-scale air quality models depend on inputs from global models to properly characterize large scale pollutant fluctuations while global models may benefit from the experiences gained in modeling air quality at finer scales motivated the organization of coordinated global and regional-scale modeling experiments under the umbrella of TF-HTAP (HTAP2) with contributions from the third phase of AQMEII (AQMEII3) as well as the MICS-Asia community as detailed in Galmarini et al. (2017). In this study, we present the results of regional-scale CMAQ simulations over North America driven by different representations of large-scale atmospheric composition as simulated by large-scale models participating in TF-HTAP. The study aims at quantifying simulated regional-scale ozone burdens both near the surface and aloft, estimating process contributions to these burdens, and calculating the sensitivity of the simulated regional-scale ozone burden to several key model inputs, in particular the global atmosphere as simulated by large-scale models and represented in CMAQ through the use of different boundary conditions. It should be noted at the outset that an intercomparison and evaluation of the various large-scale models is outside the scope of this study but is being pursued by other groups in the context of TF-HTAP.

## 2 Model Simulations and Observations

The 2010 annual simulations analyzed in this study were performed with version 5.0.2 of the CMAQ model (Byun and Schere, 2006) using meteorological fields prepared with version 3.4 of the Weather Research and Forecasting (WRF) model (Skamarock and Klemp, 2007) and emission inputs described in Pouliot et al. (2015). The CMAQ simulations were performed with a horizontal grid spacing of 12 km over the continental U.S. and used 35 vertical layers extending to 50mb.

For the base case simulations (hereafter referred to as BASE), lateral chemical boundary conditions were prepared from global concentration fields simulated by C-IFS (Flemming et al., 2015). Meteorological and air quality fields from these BASE

simulations were evaluated against observations by Solazzo et al. (2017a,b). The BASE simulations also included the tracking of contributions from different processes to ozone mixing ratios using the Integrated Process Rate (IPR) Process Analysis (PA) approach (Jeffries and Tonnesen, 1994; Jang et al., 1995) as implemented in CMAQ (Byun and Ching, 1999).

To assess the maximum impacts of boundary conditions, anthropogenic emissions within the domain, and ozone dry deposition

on simulated ozone mixing ratios, the BASE simulations were augmented by three annual bounding simulations. In the first of these bounding simulations (hereafter referred to as BC ZERO), lateral boundary conditions for all species were set to a time-invariant value of zero while all other settings were identical to BASE. In the second simulation (hereafter referred to as EM ZERO), all anthropogenic emissions as well as wildfire emissions within the domain were set to zero while all other settings were identical to BASE. For the third simulation (hereafter referred to as NO O3 DDEP), ozone dry deposition was

set to zero while all other settings were identical to BASE.

Finally, to further investigate the effects of using chemical boundary conditions derived from different global or hemispheric models, three additional annual simulations were performed using concentrations derived from: 1) CMAQ version 5.1 configured for hemispheric applications, hereafter referred to as H-CMAQ (Xing et al., 2015 a,b; Mathur et al., 2017), 2) the

GEOS-Chem model (Bey et al., 2001) version 9-01-03 which includes full tropospheric chemistry and a climatological representation of stratospheric sources and sinks, and 3) the AM3 model (Donner et al., 2011; Lin et al., 2012, 2017) with coupled stratosphere-troposphere chemistry. These simulations leveraged the coordinated AQMEII3/HTAP2 modeling experiments (Galmarini et al., 2017; Huang et al., 2017). In particular, all of these alternative global simulations providing boundary conditions (as well as the C-IFS simulations providing boundary conditions for BASE) utilized the same global

anthropogenic emission inventory described in Janssens-Maenhout et al. (2015) that is consistent with the regional-scale inventory used in the CMAQ simulations and described by Pouliot et al. (2015). However, non-anthropogenic emissions were not harmonized across the global and regional-scale simulations. As described in Flemming et al. (2015), the C-IFS simulations used lightning emissions based on the parameterization introduced in Meijer et al. (2001), biogenic emissions calculated with version 2.1 of the Model of Emissions of Gases and Aerosols from Nature (MEGAN) (Guenther et al., 2006), and biomass

burning emissions produced by the Global Fire Assimilation System (GFAS) version 1 (Kaiser et al., 2012). The H-CMAQ simulations used climatological biogenic and lightning emissions from the Global Emission Inventory Activity (GEIA) dataset (Guenther et al., 1995; Price et al., 1997) and biomass burning emissions from version 4.2 of the Emission Database for Global Atmospheric Research (EDGAR) (European Commission, 2011). The GEOS-Chem simulations used lightning emissions based on the methodology described in Murray et al. (2012), biogenic emissions calculated with MEGAN version 2.1, and

biomass burning from version 3 of the Global Fire Emissions Database (GFED) (Randerson et al., 2013; van der Werf et al., 2006). The AM3 simulations used lightning emissions based on the parameterization introduced in Horowitz et al. (2003), biogenic emissions calculated with MEGAN version 2.1, and biomass burning emissions from the Fire INventory from NCAR (FINN) (Wiedinmyer et al., 2011). The regional-scale CMAQ simulations did not include lightning emissions, calculated

biogenic emissions using version 3.14 of the Biogenic Emission Inventory System (BEIS) (Pierce et al., 1998; Vukovich et al., 2002; Schwede et al., 2005) and used 2010 wildfire emissions as described in Pouliot et al. (2015).

To create boundary conditions for the regional CMAQ simulations, outputs from the large-scale models were vertically interpolated and gas phase and aerosol species were mapped to the CB05TUCL/Aero6 mechanism used by CMAQ. Previous studies deriving regional-scale boundary conditions from global scale models noted the importance of maintaining sufficient vertical resolution in the upper troposphere / lower stratosphere in the regional model (Lin et al., 2009) and properly mapping chemical species between the modeling systems (Henderson et al., 2014). A list of the gas phase species mapped between the large-scale models and CMAQ is shown in Table 1 and a depiction of the vertical layers used in the large-scale models and regional CMAQ simulations is provided in Figure 1. Sulfate, nitrate, ammonium, elemental and organic carbon aerosols were available from all large-scale models while CMAQ trace element aerosol concentrations were estimated from large-scale model dust and sea-salt concentrations except in the case of H-CMAQ which used the same aerosol mechanism as the regional-scale CMAQ simulations. CMAQ species not available from the large scale models were obtained from the time-invariant CMAQ default profile (available at https://github.com/USEPA/CMAQ/blob/5.0.2/models/BCON/prof_data/cb05_ae6_aq/bc_profile_CB05.dat, last accessed January 10, 2018). The sensitivity simulations with the three alternate sets of boundary conditions are hereafter referred to as BC H-CMAQ, BC GEOS-Chem, and BC AM3, respectively. For reference, a list of all simulations, their acronyms and their configurations is provided in Table 2.

For the purpose of CMAQ evaluation, hourly observations of ozone were retrieved from the U.S. Environmental Protection Agency (U.S. EPA) Air Quality System (AQS) database and were used to calculate daily maximum 8-hour average (MDA8) ozone values. In addition, CASTNET hourly ozone observations were also obtained to evaluate the performance of both large-scale models and regional CMAQ at these mostly rural locations. Finally, ozonesonde observations at Trinidad Head (latitude -124.16°W, longitude 40.8°N, elevation 20m), Edmonton (latitude -114.1°W, longitude 53.55°N, elevation 766m), Churchill (latitude -94.07°W, longitude 58.75°N , elevation 30m), Boulder (latitude -105.2°W, longitude 39.95°N, elevation 1743m), Huntsville (latitude -86.59°W, longitude 35.28°N , elevation 196m), and Wallops Island (latitude -75.48°W, longitude 37.9°N, elevation 13m) were obtained from the National Oceanic and Atmospheric Administration (NOAA) Earth System Research Laboratory and the World Ozone and UV Data Center to evaluate upper air ozone simulated by the large-scale models and regional CMAQ. Model performance evaluation was performed both across the entire domain (1207 AQS monitors and 79 CASTNET monitors) and separately for five sub-regions that are characterized by differences in their proximity to the domain boundaries, elevation, and relative abundance of anthropogenic and biogenic emissions: Northwest (NW) (41 AQS monitors and 2 CASTNET monitors), Intermountain West (IMW) (53 AQS monitors and 7 CASTNET monitors), Midwest (MW) (195 AQS monitors and 13 CASTNET monitors), Southeast (SE) (166 AQS monitors and 13 CASTNET monitors), and Northeast (NE) (204 AQS monitors and 15 CASTNET monitors). Note that these analysis sub-regions do not cover the entire modeling

domain. For all comparisons of observations and model simulations presented in this study, data pairs were included in the computation of derived metrics, such as daytime averages (defined as average mixing ratios between 10 am and 5 pm local standard time) or monthly averages, only when both observations and model simulations were available for a given hour. Furthermore, each monitored value was paired with the corresponding model value based on the model grid cell in which the

monitor was located. In particular, multiple observations within the same grid cells were not averaged because the definition of the horizontal grids varied between all the simulations analyzed in this study. For seasonal analyses, winter was defined as December – February, spring was defined as March – May, summer was defined as June – August, and fall was defined as September – November. Figure 2 shows a map of the entire WRF/CMAQ 12km modeling domain, these five analysis regions, and the location of the AQS monitors, CASTNET monitors, and ozonesonde sites used in the analysis.

**3 Results and Discussion**

**3.1 Analysis of BASE CMAQ Simulations**

**3.1.1 Evaluation Summary**

Before analyzing ozone results for the sensitivity simulations, this section provides an overview of ozone model performance in the five analysis regions used in this study for the BASE simulation. Results for meteorology, ozone and other pollutants

from these simulations were already analyzed and compared to other models in Solazzo et al. (2017 a,b).

Table 3 provides a summary of model performance for the BASE simulation for MDA8 ozone at AQS monitors over the five analysis regions shown in Figure 2. The metrics shown in this table are the Normalized Mean Bias (NMB), Normalized Mean Error (NME), and correlation coefficient (R). These metrics were computed at each site for each season and the median metric

across all sites in a given region and season is shown in Table 3. The cells in the table are color-coded based on the model evaluation goals and acceptability criteria proposed by Emery et al. (2017) based on a review of published model evaluation studies. Green cells indicate regions and seasons where model performance meets the goal for a given metric (NMB <+- 5%, NME <15%, and R > 0.75), yellow cells indicate regions and seasons where model performance meets the acceptability criterion but not the goal (+-5% < NMB <+- 15%, 15% < NME <25%, and 0.5 < R < 0.75), and orange cells indicate regions

and seasons where neither the goal nor the acceptability criterion are met (NMB >= +/- 15%, NME >= 25%, R <= 0.5). Regionally, results show that model performance tends to be worst in the NW compared to other regions while seasonally, model performance tends to be worst during winter compared to other seasons. The three instances of model performance not meeting the acceptability criterion proposed by Emery et al. (2017) all occur during the winter. Except for the NW, NMB is negative during winter in all regions, suggesting that large-scale ozone background concentrations specified through C-IFS

provided model boundary conditions may be underestimated in this simulation, particularly over the Northern portion of the modeling domain. This is consistent with the findings of Flemming et al. (2017).

The model performance overview presented in Table 3 does not provide information on the ability of the BASE simulation to capture different portions of the observed MDA8 ozone distribution. To this end, we also computed differences between observed and modeled MDA8 ozone distributions at AQS monitors for each season and analysis region. For each season and region, the observed MDA8 ozone concentrations were rank ordered at each station. Differences between CMAQ simulations and observations were then computed for each observed percentile either by selecting the model value corresponding to the date of the observed percentile (paired-in-time comparison) or rank-ordering the model values and then selecting the modeled percentile corresponding to the observed percentile (unpaired-in-time comparison). The median value of these paired-in-time and unpaired-in-time differences across all AQS stations in a given season and region was then computed for each observed percentile and is shown in Figure 3.

One general feature visible throughout all seasons and regions is that the unpaired-in-time differences tend to be more flat across the range of the observed percentiles while the curves for the paired-in-time differences tend to have a negative slope. This different behavior of the unpaired-in-time and paired-in-time comparison indicates that the CMAQ simulations have better skill in capturing the width of the observed MDA8 distribution than in capturing the timing of specific observed ozone events. The NW is the only region with positive unpaired-in-time differences throughout all seasons. The IMW has the least spread in model performance across seasons for all percentiles, both in terms for unpaired-in-time and paired-in-time differences. Unpaired-in-time winter results for the MW, SE, and NE show an underestimation of observed MDA8 ozone across all percentiles. This is also true for the comparison of paired-in-time differences for all observed percentiles greater than the 20[th] percentile. In contrast, summer differences in these regions tend to be positive for all but the highest percentiles. For all regions, model-observations differences for spring and fall tend to be similar to each other. For the MW, SE, and NE regions, differences for these seasons fall between the winter and summer results with consistently small unpaired-in-time differences and a tendency to overestimate lower observed percentiles and underestimate higher observed percentiles when considering paired-in-time differences. The analysis presented in Section 3.2.3 will explore the sensitivity of these model performance results towards alternate lateral boundary conditions.

### 3.1.2 Process Analysis Contributions to Ozone Columns

The analysis above focused on ground-level ozone evaluation. Ground level ozone is affected by a number of physical and chemical processes both near the surface and aloft. The PA tool in CMAQ (Jeffries and Tonnesen, 1994; Jang et al., 1995) provides a method to track these process contributions to the modeled ozone. In this study, we configured PA to track the contributions of the following processes to simulated ozone: horizontal advection (HADV), vertical advection (ZADV), horizontal diffusion (HDIF), vertical diffusion (VDIF), dry deposition (DDEP), chemistry (CHEM), and cloud processes including vertical mixing by convective clouds and removal through scavenging and aqueous chemistry (CLDS). The resulting process contributions are available for each grid cell and each hour throughout the annual simulation.

Figure 4 shows profiles of seasonal total ozone column mass changes for each model layer due to the seven processes summed over the entire modeling domain. The PA terms represent the net change in ozone mass due to a given process in a given model layer and season. For almost all layers and seasons, HADV and ZADV are of similar magnitude and opposite direction due to mass consistent advection and are the dominant processes for layers above ~800 mb. In the first model layer, DDEP is a strong sink of ozone, balanced largely by VDIF, i.e. flux of ozone from upper layers to the surface. VDIF tends to become insignificant above ~500 mb while the effect of HDIF is negligible for all model layers. CHEM is a sink in the first model layer, a source in the boundary layer, and a net sink between approximately 800 mb and 400 mb for all seasons except winter. CLDS tends to be a source of ozone in the lower atmosphere and a sink in the upper atmosphere.

To better illustrate the seasonal variations of the process contributions in the upper model layers, free troposphere, and boundary layer / lower troposphere, Figure 5 presents monthly domain-wide total PA contributions to ozone columns in CMAQ layers 1-21 (surface to approximately 750 mb), 22 – 31 (approximately 750 – 250 mb), and 32 – 35 (approximately 250 mb – 50 mb). The horizontal and vertical advection and diffusion terms were summed to compute the effects of total advection (TADV) and total diffusion (TDIF), respectively. Consistent with the profiles shown in Figure 4, changes in ozone mass in the upper layers are dominated by TADV, with these layers gaining ozone mass through TADV early and late in the year when tropopause heights are lower and a larger portion of the lower stratosphere is included in the model while they tend to lose mass through the effects of TADV from April through September. The column between 250 mb and 750 mb gains ozone mass through TADV for almost all months, indicating that both lateral boundary conditions and ozone in the upper layers determine the ozone column burden simulated in the free troposphere. CHEM is a net sink especially during summer. Vertical mixing by convective clouds also removes ozone from these layers while the effect of TDIF is small. The ozone column below 750 mb gains mass through the effects of CHEM especially during summer as well as through the effects of vertical mixing by convective clouds that tap into the ozone reservoir in the free troposphere to enhance the lower atmospheric ozone burden. The dominant sink term of ozone mass in this layer range is DDEP at the surface. TADV and TDIF play a secondary role in modifying the total ozone burden in this column range. It should be noted that the PA results shown in Figures 4-5 are based on a single year. Inter-annual variability would be expected to affect the absolute magnitude and month-to-month variations especially of the advection processes, however, the qualitative differences in process rankings between different layer ranges would be expected to be robust with respect to inter-annual variability. Moreover, the process contributions presented here are monthly totals over the entire domain. Contributions for specific locations and episodes would likely differ. For example, while the CLDS term is shown to be a net source for lower tropospheric ozone over the entire domain, it might be a net sink during episodes of high ozone formation in the boundary layer. Overall, the results indicate that alternate model representation of advection, dry deposition and cloud processes as well as alternate model inputs (boundary conditions affecting advected ozone and emissions affecting ozone chemistry) would be expected to have noticeable effects on the simulated ozone burdens

and their seasonal variation. Hypothetical bounding scenarios quantifying the effects of emissions, boundary conditions, and ozone dry deposition on the simulated ozone burden are explored in the next section.

### 3.1.3 Brute Force Bounding Simulations

The upper three panels of Figure 6 present time series of the monthly average ozone column mass for the BASE, BC ZERO,
EM ZERO, and NO O3 DDEP sensitivity simulations for the same three layer ranges analyzed in the previous section while the lowest panel presents time series of monthly average ozone mixing ratios for the first model layer. These time series confirm the PA findings that the ozone column burden above 250 mb is almost entirely driven by advection of lateral boundary conditions in these continental-scale CMAQ simulations. Specifically, in this layer range the BC ZERO simulation has an ozone column of essentially zero while the burdens simulated for the EM ZERO and NO O3 DDEP cases are indistinguishable
from the burden simulated for the BASE case. The results for the free troposphere (750 mb – 250 mb) show a small difference in the column base simulated by the BASE and EM ZERO simulations especially during summer. This difference quantifies the net effects of ozone production from emissions but is dwarfed by the impacts from the BC ZERO simulation which again is suggestive that the variability in the free troposphere is largely driven by the specification of lateral boundary conditions. Results for the column from the surface to 750 mb show noticeable differences in ozone column mass between all four
simulations, with the differences with respect to the BASE simulation being lowest for the NO O3 DDEP case and highest for the BC ZERO case. For the surface ozone monthly mean mixing ratio, the largest signal is seen for the NO O3 DDEP case followed by the BC ZERO case. The EM ZERO case has the smallest impact at the surface but, as shown above, emissions have a larger cumulative impact on column ozone burden than dry deposition.

Furthermore, the surface results for the BASE, BC ZERO and EM ZERO sensitivity simulations indicate that during wintertime, domain-average simulated ozone mixing ratio are almost exclusively driven by boundary conditions, i.e. the BASE and EM ZERO are very similar despite the lack of anthropogenic emissions in the latter, and mixing ratios in the BC ZERO simulation are close to 0 ppb. The EM ZERO results also indicate that the impact of boundary conditions on regional ozone is largest in springtime when free tropospheric ozone in the northern hemisphere reaches a maximum. If one views the ozone
from the EM ZERO simulation as the amount of regional ozone due to boundary conditions and biogenic emissions, and BC ZERO as the amount of ozone due to anthropogenic and biogenic emissions within the domain, the results indicate that the former dominates the latter throughout the year in terms of domain-average monthly mean mixing ratios at the surface. However, it should be noted that the impacts of these bounding simulations on simulated surface ozone vary spatially. Solazzo et al. (2017b) analyzed seasonal cycles from these simulations sampled at ozone monitoring locations and found that during
the summer time the impact of anthropogenic emissions on monthly mean concentrations was comparable to or larger than the impact of boundary conditions in the subregions considered in their analysis.

To investigate the spatial variability of surface ozone from these bounding scenarios, Figure 7 shows maps of differences in seasonal mean mixing ratios between the three sensitivity simulations (BC ZERO, EM ZERO, and NO O3 DDEP) and the BASE simulation. The results show that as expected the impact of zeroing out boundary conditions decreases with distance from the boundaries in all seasons, with the smallest impacts typically seen in the southeastern portion of the modeling domain.

In contrast, the effects of zeroing out the anthropogenic and wildfire emissions tend to be largest in the eastern portion of the modeling domain, leading to larger decreases in simulated ozone compared to the BC ZERO case during summer in that region. Increases of seasonal mean ozone can be observed in urban areas for the EM ZERO simulation in all seasons. The effects of ozone dry deposition on simulated seasonal mean surface ozone mixing ratios is most pronounced in the eastern portion of the modeling domain during spring and especially summer, with increases of more than 20 ppb simulated across a broad region. These NO O3 DDEP results indicate that intercomparing and evaluating ozone dry deposition approaches would be a fruitful avenue for future model intercomparison activities aimed at better constraining processes affecting surface ozone fluctuations simulated by different models.

Overall, the analysis of the brute-force sensitivity simulations presented in this section as well as the process analysis results presented in Section 3.1.2 confirm that the characterization of ozone outside the regional-scale modeling domain can have a profound impact on simulated regional-scale ozone. However, these brute force bounding simulations do not represent plausible representations of real-world conditions. In the next section, we present regional-scale CMAQ simulations utilizing boundary conditions derived from different large-scale models. This is aimed at investigating the impact of different state-of-science representations of the global atmosphere on air quality simulated over the United States with a 12 km resolution regional-scale model.

## 3.2 Analysis of CMAQ Simulations with Boundary Conditions from Different Global Models

### 3.2.1 Comparisons of Aloft Concentrations from Global Models and Regional CMAQ

Figure S1 shows time-height cross sections of monthly mean ozone mixing ratios along the western, southern, eastern, and northern boundary of the regional CMAQ domain for the four large-scale models from which boundary conditions were derived. The mixing ratios were averaged over all columns or rows defining a given boundary and also were averaged for each month. For all boundaries, GEOS-Chem and C-IFS tend to have the highest ozone mixing ratios for levels above 150 mb. All models show a springtime maximum and fall minimum for these levels. During springtime, AM3 shows the deepest intrusion of higher ozone mixing ratios from upper levels to mid- and lower-tropospheric levels at the western, northern and eastern boundary.

Time-height cross sections of monthly mean ozone were also prepared at the location of the six ozonesonde stations shown in Figure 2. These monthly mean mixing ratios were calculated for observations, the four large-scale models, and the corresponding four regional CMAQ simulations. Since ozonesonde measurements are available at a much higher vertical

resolution than the model simulations, observations were vertically averaged to the vertical structure used by each model (see Figure 1) and the observations as averaged to the C-IFS layer structure are depicted in Figures 8a-b. Note that even though observations and large-scale model predictions (except H-CMAQ) are available for higher altitudes (see Figure 1), only values up to the highest model level below 50 mb were extracted for these figures to be comparable to the output from the regional-scale CMAQ simulations (specifically, C-IFS values were only extracted up to layer 38, GEOS-Chem values were only extracted up to layer 37, and AM3 values were only extracted up to layer 26 for this comparison). For easier comparison between models and sites, all figures use a common vertical pressure range of 1025 mb to 50 mb even though this full range is not covered at all sites and by all models. Figure 8a shows the time-height cross sections for the three ozonesonde sites that are located in close proximity of the western and northern regional CMAQ boundaries (i.e. Trinidad Head, Edmonton, and Churchill) where inflow into CMAQ is expected to be most important due to prevailing flow patterns. The cross sections for the large-scale models in rows 1, 3 and 5 are consistent with the cross sections for the western and northern boundaries shown in Figure S1. In particular, GEOS-Chem and C-IFS tend to have the highest ozone mixing ratios for levels above 150 mb while AM3 shows the deepest intrusion of higher ozone mixing ratios from upper levels to mid- and lower-tropospheric levels especially during springtime. Comparing the large-scale model results to the observed cross sections in the left column reveals that free tropospheric mixing ratios simulated by C-IFS, H-CMAQ and GEOS-Chem tend to be closer to the observations than the mixing ratios simulated by AM3 which tend to be overestimated. Another key feature of the cross sections shown in Figure 8a is that the regional CMAQ results at Trinidad Head and Edmonton shown in rows 2 and 4 closely mirror those simulated by the corresponding large-scale models, emphasizing the impact of boundary conditions on regional-scale simulations especially near the boundaries (note that no regional-scale results are shown for Churchill as the station is located outside the 12 km modeling domain shown in Figure 2).

Figure 8b shows corresponding results for the three ozonesonde locations in the interior of the regional-scale CMAQ modeling domain: Boulder, Huntsville, and Wallops Island. At all of these sites, AM3 tends to simulate higher free tropospheric and lower tropospheric mixing ratios than the other large-scale models during spring while GEOS-CHEM tends to simulate higher mixing ratios during summer. The observed cross section at Boulder suggests that no large-scale model performs systematically better or worse than another at this location in the free troposphere. At Huntsville and Wallops Island, free tropospheric mixing ratios are overestimated by AM3 during spring and by GEOS-CHEM during summer. Finally, the comparison between the regional CMAQ cross sections and the corresponding large-scale model cross sections at these three sites shows some differences as well as similarities, indicating that differences in factors such as the treatment of vertical mixing, lightning emissions, chemistry, deposition and biogenic emissions can lead to deviations between the large-scale models and the regional CMAQ simulations over the continental United States.

The connection between large-scale models and the corresponding regional CMAQ simulations is further explored in Figure 9. This figure shows monthly average time series of 500 mb observed ozone, ozone simulated by the large-scale models (solid

lines), and ozone simulated by regional CMAQ driven with boundary conditions from the different large scale models (dashed lines) at Trinidad Head, Edmonton, Boulder, Huntsville, and Wallops Island. Model simulations were extracted for the layer closest to 500 mb and observations were vertically averaged across the depth of each of these different model layers. As a result of the different vertical structure of the four large-scale models and the regional CMAQ simulations depicted in Figure

1, five different estimates of 500 mb observations were derived and the range of these different estimates is indicated by the shaded area in Figure 9. Between March and June, AM3 mixing ratios are up to 20 ppb higher than the mixing ratios simulated by the other three large-scale models at Trinidad Head, Boulder and Wallops Island and up to 10 ppb higher at Boulder and Huntsville. At all sites except Boulder, the AM3 simulations are also systematically higher than observations during this time period. At the sites closest to the western and northern inflow boundaries, i.e. Trinidad Head and Edmonton, the time series

for the regional CMAQ results closely mirror those for the corresponding large-scale models. Within the modeling domain, there is more separation of the large-scale and regional CMAQ results, especially between the GEOS-Chem and BC GEOS-Chem results during summer at Huntsville and Wallops Island where BC GEOS-Chem simulates substantially lower mixing ratios than GEOS-Chem. These differences may be at least partially due to the representation of emissions from lightning. While the regional CMAQ simulations did not include lightning NO emissions, it was included in the GEOS-Chem

simulations. Zhang et al. (2014) and Travis et al. (2016) note that the standard GEOS-Chem treatment of lightning NOx yields for midlatitudes may be too high and can lead to positive ozone biases at the surface.

The differences in the magnitude of mid-tropospheric ozone mixing ratios between the large-scale models at the more remote Trinidad Head, Edmonton, and Churchill sites point to differences in the representation of stratospheric ozone and

20 stratosphere/troposphere exchange processes. The representation of the latter might also be affected by differences in vertical resolution as shown in Figure 1. In conjunction with the results presented in Section 3.1, Figures 8 and 9 also suggest that regional scale CMAQ simulations using these four different sets of boundary conditions will yield different estimated ozone burdens. It should be noted that in-depth evaluation and intercomparison of the different large-scale simulations is beyond the scope of the current study. Previous studies evaluating H-CMAQ, GEOS-Chem, C-IFS and AM3 include Xing et al. (2015a,b),

Mathur et al., (2017), Fiore et al. (2009), Flemming et al., (2015), and Lin et al. (2012a,b, 2017). Three of these simulations (GEOS-Chem, C-IFS and AM3) are also being compared against aloft and surface ozone measurements by Cooper et al. (2017).

### 3.2.2 Seasonal Differences in CMAQ Simulated Ozone Columns

Figure S2 shows daily time series of CMAQ-simulated domain-total ozone column mass for the same three layer ranges used

in the previous sections. The results are for the BASE, BC H-CMAQ, BC GEOS-Chem, and BC-AM3 simulations. For layers 32 – 35 (i.e. the layers approximately above 250 mb), all simulations show a maximum in spring and a minimum in fall. All simulations track each other but the magnitudes differ by up to a factor of two. The largest ozone abundance in the upper layers is simulated by BC GEOS-Chem, followed by BASE, BC AM3 and BC H-CMAQ, consistent with the analysis of boundary

conditions in the Section 3.2.1. The most notable feature for the ozone column mass in layers 22 – 31 (i.e. approximately 750 mb – 250 mb) is the larger springtime ozone burden simulated by BC AM3 compared to the other three simulations, consistent with the analysis of the ozone boundary conditions at 500 mb in the previous section. The same feature is also found for the ozone column mass in layers 1- 21 (i.e. surface to approximately 750 mb) which confirms the notion that vertical exchange
between this layer range and the free troposphere leads to a tight coupling of their ozone fluctuations. For all layer ranges, these results confirm that differences in ozone boundary conditions result in differences of CMAQ-simulated ozone column mass over the modeling domain.

### 3.2.3 Seasonal Differences in CMAQ Surface Ozone Mixing Ratios

Figure 10 shows maps of seasonal mean ozone mixing ratios at the surface for the four simulations. The left column shows the
mixing ratios for the BASE simulation while the second, third and fourth columns show the differences between BC H-CMAQ and BASE, BC GEOS-Chem and BASE, and BC AM3 and BASE, respectively.  For the BASE simulations, many regions including the intermountain west and the central U.S. show a springtime peak in seasonal mean ozone while summer peaks are present downwind of more urban areas such as in California and the Mid-Atlantic corridor. Differences between the BASE simulations and the three sensitivity simulations are generally highest near the domain boundaries in all seasons but differences
of 10 ppb in seasonal mean $O_3$ can be found even in the center of the modeling domain in some cases. The largest differences exist between the BC AM3 and BASE simulations and are especially pronounced during spring and winter. In contrast, the differences between BC H-CMAQ and BASE and BC GEOS-Chem and BASE are typically smaller (+/- 4 ppb for most of the modeling domain except for BC H-CMAQ during winter). These impacts of lateral boundary conditions on surface ozone mixing ratios are consistent with the analysis of the large-scale models and CMAQ ozone column burdens in the previous
sections. Separate analysis shows that considering MDA8 ozone instead of hourly ozone leads to very similar spatial patterns of seasonal mean differences between the model simulations. This is expected since the effect of boundary conditions on the average diurnal cycle manifests itself mostly as a constant shift throughout the course of the day as shown in Solazzo et al. (2017b).

Figure 11 a-f shows time series of differences between modeled and observed ozone mixing ratios. Panel a) shows results for monthly means of daytime average mixing ratios at CASTNET monitors for the four regional model simulations, panel b) shows the results for the four corresponding large-scale models, panel c) shows results for monthly means of daytime average mixing ratios at AQS monitors instead of CASTNET monitor for the four regional model simulations, panel d) shows results for the four regional models at AQS monitors using monthly means of MDA8 instead of monthly means of daytime average
mixing ratios, and panels e) and f) correspond to panels c) and d) but show daily rather than monthly mean values. These time series illustrate that regardless of metric (daytime average vs MDA8) and network (CASTNET vs. AQS), all regional CMAQ simulations overestimate domain-mean observed ozone throughout the year with the exception of the BASE simulation during winter, with the overestimation being most pronounced for BC AM3 during spring. The spread in monthly MDA8 ozone biases

(i.e. model minus observation differences) between the four regional CMAQ simulations is on the order of 7-10 ppb for most of the year at AQS sites, i.e. roughly 15% - 30% of simulated monthly mean values. The spread is smaller from June to September when it drops to less than 5 ppb. The spread in biases of domain-wide daily MDA8 ozone at AQS sites can reach as high as 15 ppb during springtime. In contrast to the comparison of regional CMAQ and large-scale model results for aloft

ozone in Section 3.2.2, the comparison of panels a) and b) shows that model performance for daytime average surface ozone mixing ratios at CASTNET monitors are not tightly linked between these two groups of simulations. This again indicates that while CMAQ free tropospheric ozone mixing ratios are dominated by advection, other factors modulate surface ozone, including the treatment of vertical mixing, chemistry, deposition and biogenic emissions. Moreover, the larger spread in model bias for the large-scale models compared to regional CMAQ can be explained by the fact that the large scale models differ in

their representation of many of these processes while the four regional CMAQ simulations share all input files and process representations and only differ in their representation of large-scale background concentrations. The comparison of panels a) and b) also illustrates that the biases of the regional CMAQ simulations are comparable to or lower than the biases of the large-scale models.

The bias time series in Figure 11 considered spatial averages over all CASTNET or AQS monitors. To investigate spatial variations in these biases, Figure 12 shows maps of seasonal mean biases for daytime average ozone at CASTNET sites for BASE, BC H-CMAQ, BC GEOS-Chem, and BC AM3 (rows 1 and 3) and C-IFS, H-CMAQ, GEOS-Chem, and AM3 (rows 2 and 4) for spring (rows 1 and 2) and summer (rows 3 and 4). These maps correspond to the time series shown in Figures 11a)-b). Two features stand out in these maps. First, all regional CMAQ simulations and corresponding large-scale simulations tend

to be positively biased in the Eastern United States during spring and summer; this is especially pronounced for C-IFS, GEOS-Chem and AM3 during summer. Significant positive ozone biases at CASTNET sites in the SE were also reported for GEOS-Chem for summer 2013 by Travis et al. (2016) who attributed a large portion of the bias to overestimated anthropogenic $NO_x$ emissions. In the current study, the annual total anthropogenic $NO_x$ emissions are shared across all regional and large-scale simulations since the HTAP2 global inventory (Janssens-Maenhout et al., 2015) incorporated the AQMEII2 regional inventory

(Pouliot et al., 2015) over North America, although differences may exist in terms of temporally and vertically allocating these emissions for a specific model. This suggest that factors other than anthropogenic emissions, such as biogenic emissions, chemistry, and deposition that differ between the large-scale models as well as between the large-scale models and regional CMAQ also affect the ozone bias in this region. Second, consistent with the time series shown in Figures 11a)-b), the model performance of the regional CMAQ simulation and the corresponding large-scale simulation are not tightly linked. As

discussed above, this indicates that while free tropospheric regional CMAQ ozone mixing ratios are dominated by advection, other factors including the treatment of vertical mixing, chemistry, deposition and biogenic emissions modify surface ozone. However, despite the general differences between the regional CMAQ and large-scale model results, the bias patterns during spring and summer tend to be most similar between BC H-CMAQ and H-CMAQ compared to all other pairs of regional / large-scale models, likely pointing to greater consistency in the treatment of physical and chemical processes across scales for

this particular combination. It should be emphasized that the comparison of regional and large-scale model biases in Figures 11a)-b) and 12 is not aimed at establishing the relative merits of either modeling approach or of using one set of boundary conditions over another in the regional CMAQ simulations but rather to illustrate the magnitude of the impact of modeling choices on model performance.

Table 4 a)-c) presents corresponding model performance metrics (NMB, NME, R) for MDA8 $O_3$ at AQS monitors across the five analysis regions and four seasons for the four CMAQ simulations with different boundary conditions. Consistent with the results for the BASE simulations evaluated in Table 3 in Section 3.1.1, model performance for all simulations tends to be worst in the NW. However, the noteworthy feature of the results shown in Tables 4 a-c is that boundary conditions can have a

10 substantial impact on model performance as measured by the goals and acceptability criteria proposed by Emery et al. (2017). Boundary conditions also can affect conclusions about the directionality of the model bias. While wintertime MDA8 $O_3$ is underestimated by the BASE run for all regions except NW as shown earlier, the opposite is true for the BC H-CMAQ and BC AM3 simulations. Regardless of whether or not these proposed model performance acceptability criteria will ultimately be adopted by the regional air quality model community, the results presented here show that the choice of lateral boundary

conditions would be influential in measuring model performance against these acceptability criteria.

The results above assess the impact of different boundary conditions on model performance as measured across an entire season. Figure 13 shows paired-in-time CMAQ-observation differences of MDA8 $O_3$ at AQS monitors across the range of observed percentiles for each simulation, season, and region, analogous to the results shown in Figure 3 for the BASE

simulations. Overall, these graphs indicate that boundary conditions can affect model performance across the entire range of the observed distribution, although the impacts tend to be lower during summer and for the very highest observed percentiles. The results also reaffirm that the differences between the four simulations tend to be largest during winter and spring across all regions. During spring, most of the spread is caused by the higher MDA8 ozone values simulated by BC AM3 compared to the other three simulations across all regions. During summer, BC AM3 results are noticeably higher than results from the

other three simulations only over the NW and NE regions. During fall, this is the case only for the NW region, while for the other four regions there is roughly equal spread between all simulations for all percentiles. During winter when local production is small, the difference in lateral boundary conditions results in a clear separation between the four simulations across all regions and percentiles.

Corresponding paired-in-time results comparing daytime average $O_3$ from the large-scale models and corresponding regional CMAQ simulations against observations at CASTNET monitors are presented in Figure S3 and S4. The daytime average CMAQ results at CASTNET monitors in Figure S4 are very similar to the MDA8 ozone results at AQS monitors shown in Figure 13, consistent with the comparison of difference time series for different metrics and networks in Figure 11. The spread in model-observation differences is larger for the large-scale models than the spread for the regional CMAQ results for most

percentiles, seasons and regions (note that the y-axis range for the large-scale model results in Figure S3 is larger than the range for the CMAQ model results in Figure S4). In contrast to the CMAQ results which show a similar relative ranking of the four simulations for most seasons and regions (with BC AM3 generally having the highest model-to-observation differences, followed by BC H-CMAQ, BC GEOS-Chem, and BASE), the performance of the four large-scale models shows more variable behavior with no clear and systematic model-to-model differences across seasons and regions. This reaffirms that differences in free tropospheric ozone at the boundary of regional simulations can have a systematic impact on such regional simulations throughout the domain while the effects of other model differences (e.g. transport, vertical mixing, chemistry and deposition) manifest themselves in a spatially and temporally more complex manner. As a result, there is no clear similarity between surface ozone model performance for the large-scale models and the performance of the regional CMAQ simulations with the possible exception of the H-CMAQ / BC H-CMAQ pair which shares process representations across scales.

## 4 Summary and Discussion

The results presented in this study are aimed at quantifying CMAQ-simulated regional-scale ozone burdens both near the surface and aloft, estimating process contributions to these burdens, and calculating the sensitivity of the simulated regional-scale ozone burden to several key model inputs with a particular emphasis on boundary conditions. The model simulations supporting this analysis were performed over the continental U.S. for the year 2010 within the context of the AQMEII3/HTAP2 activities. Process analysis was employed to track the contributions of horizontal and vertical advection and diffusion, dry deposition, chemistry and cloud processes on simulated ozone burdens. Changes in ozone mass in the upper layers were found to be dominated by advection. Advection also is the largest source of ozone for the column between 250 mb and 750 mb throughout most of the year, indicating that both lateral boundary conditions for this layer range and ozone in the upper layers (which in turn depends on lateral boundary conditions specified for the upper layers) have a profound impact on the burden simulated in the free troposphere. Chemistry and vertical mixing by convective clouds are the main sink for this column range. The ozone column below 750 mb gains mass through the effects of chemistry especially during summer as well as through the effects of vertical mixing by convective clouds that tap into the ozone reservoir in the free troposphere to enhance the lower atmospheric ozone burden. The dominant sink term of ozone mass in this layer range is dry deposition at the surface. Advection and diffusion play a secondary role in modifying the domain-total ozone burden in this column range. These PA contributions to CMAQ simulated ozone column burdens indicate that alternate model representation of advection, dry deposition and cloud processes as well as alternate model inputs (boundary conditions affecting advected ozone and emissions affecting ozone chemistry) would be expected to have noticeable effects on the simulated ozone burdens and their seasonal variation.

Hypothetical bounding scenarios were performed to quantify the effects of emissions, boundary conditions, and ozone dry deposition on the simulated ozone burden by zeroing out each of these factors in turn. Analysis of these simulations confirmed

the key importance of boundary conditions which dominates over the other two factors for the free and upper troposphere and lower stratosphere. Ozone burdens below 750 mb and especially ozone mixing ratios at the surface show significant changes in the no emissions and no ozone dry deposition simulations, and the relative impact of all three bounding simulations on surface ozone varies seasonally and spatially. Overall, the analysis of the brute-force sensitivity simulations confirms that the characterization of ozone outside the regional-scale modeling domain can have a profound impact on simulated regional-scale ozone.

Four global and hemispheric modeling systems, i.e. C-IFS, H-CMAQ, GEOS-Chem, and AM3, were used to derive alternate boundary conditions for the regional-scale CMAQ simulations. When comparing ozone from these four large-scale models against each other along the boundaries of the regional-scale CMAQ domain, noticeable differences were found both in terms of the magnitude and seasonal variations of ozone mixing ratios. GEOS-Chem and C-IFS simulated the highest ozone mixing ratio in the stratosphere while AM3 generally simulated the largest ozone mixing ratio in the free troposphere and PBL. Model-to-model differences in the magnitude and seasonal variations of ozone mixing ratios along the regional model boundaries in the mid-troposphere point to differences in the representation of stratospheric ozone and stratosphere/troposphere exchange processes in the large-scale models.

The regional-scale CMAQ simulations using these four different boundary conditions showed that the largest ozone abundance in the upper layers was simulated by BC GEOS-Chem, followed by BASE (using C-IFS lateral boundary conditions), BC AM3 and BC H-CMAQ, consistent with the analysis of the ozone fields from the large-scale models along the CMAQ boundaries and with the notion that the stratospheric ozone burden simulated by regional-scale CMAQ is driven by advection of lateral boundary conditions. The most notable feature for the ozone column mass in the mid-troposphere was found to be the larger springtime ozone burden simulated by BC AM3 compared to the other three simulations, again consistent with the analysis of the ozone boundary conditions in that layer range. The same feature was also found for the ozone column mass closer to the surface which confirms the notion that vertical exchange between this layer range and the free troposphere leads to a tight coupling of their ozone fluctuations. For all layer ranges, the analysis of these regional-scale CMAQ simulations highlighted that differences in ozone boundary conditions result in differences of CMAQ-simulated ozone column mass over the modeling domain.

The results for surface ozone mixing ratios are consistent with the results for the free tropospheric and lower tropospheric / PBL ozone burdens. In particular, the largest differences between the four sets of simulations exist between the BC AM3 and BASE simulations and are especially pronounced during spring and winter where they can reach more than 10 ppb for seasonal mean ozone mixing ratios and as much as 15 ppb for domain-averaged MDA8 ozone on individual days. In contrast, the differences between BC H-CMAQ and BASE and BC GEOS-Chem and BASE are typically smaller (+/- 4 ppb for most of the modeling domain except for BC H-CMAQ during winter). Comparing simulated surface ozone mixing ratios to observations

and computing seasonal and regional model performance statistics revealed that boundary conditions can have a substantial impact on model performance and can also affect conclusions on the directionality of model biases. Further analysis showed that boundary conditions can affect model performance across the entire range of the observed distribution, although the impacts tend to be lower during summer and for the very highest observed percentiles.

While the results presented in this manuscript highlight the importance of boundary conditions for regional-scale ozone simulations, it should be noted that they were based on a single year of simulations. Many previous studies have shown a strong connection between inter-annual meteorological variability and ozone on continental- to global scales (Lin et al., 2012a,b, 2017; Hegarty et al., 2007; Porter et al., 2017; Hogrefe et al., 2011), especially as it relates to the impacts of variations

in hemispheric-scale ozone on regional-scale ozone. Future work analyzing multi-year simulations from multiple global models linked to corresponding regional-scale simulations would be beneficial in better constraining the effects of large-scale inter-annual variability on simulated regional-scale ozone burdens and the inter-annual variability of contributions from large-scale ozone to surface ozone especially during time periods of elevated concentrations.

The results shown in Section 3 (e.g. Figures 8-9) strongly suggest that differences in the mid-tropospheric ozone mixing ratios simulated by the large-scale models were the main driver of ozone differences between the corresponding regional-scale CMAQ simulations. However, differences in other species such as PAN, differences in the availability of a complete set of CMAQ species from all large-scale models (see Table 1), and inconsistencies in chemical speciation between the large-scale models and regional-scale CMAQ may also have contributed to the ozone differences between the regional-scale CMAQ

simulations. Thus, while linking output from available global or hemispheric models to regional-scale models despite such differences represents current best practices in the regional-scale air quality modeling community, additional research should be geared towards developing modeling frameworks that enable a consistent representation of model processes, species, and vertical grid representation across scales. An example of such efforts is the ongoing work to extend CMAQ to hemispheric scales (Mathur et al., 2017). Ensuring such consistency does not in itself guarantee improved model performance but would

allow for more targeted diagnostic model evaluation aimed at specific processes which is more challenging when linking together different modeling systems. To achieve such consistency, future work should also be directed toward developing and implementing scale-dependent treatment for atmospheric chemistry in next-generation global dynamic models with variable grid resolution features such as the Model for Prediction Across Scales (MPAS) (Skamarock et al., 2012) and the Finite-Volume Cubed-Sphere Dynamical Core (FV3) model (Harris and Lin, 2013). Finally, the results from the bounding

sensitivity simulations suggest that coordinated evaluation and intercomparison activities for ozone dry deposition would be valuable in better constraining simulated ozone budgets.

In addition to these potential future research directions for the global and regional-scale air quality modeling communities, there are also several more concrete opportunities for further analyses that could be pursued as part of the current collaboration

between AQMEII and TF-HTAP. First, while the present study shows that different boundary conditions can have an impact across the entire range of modeled ozone mixing ratios, it does not analyze such impacts during specific events and at specific locations. Such case study analyses could be the topic of future work. Second, the CMAQ PA results indicate the importance of vertical mixing processes (including mixing by convective clouds), advection and dry deposition on the modeled vertical

distribution of ozone. Inert tracers of boundary conditions included in the AQMEII3 simulations analyzed by Solazzo et al. (2017a) can aid in the diagnosis of how model-to-model differences in these processes affect the impact of boundary conditions on ozone simulated by different regional-scale models (Liu et al., 2017). Finally, the "EM ZERO" bounding simulation could be further analyzed in the context of estimating "North American Background" (NAB) (Fiore et al., 2014) or "U.S. Background" (USB) (Dolwick et al., 2015) ozone, especially if this bounding simulation were to be repeated with lateral

boundary conditions derived from H-CMAQ, GEOS-Chem, and AM3 instead of C-IFS. However, even without such additional runs, the results from the simulations with different boundary conditions performed for base emission conditions suggest that estimated NAB or USB values resulting from such potential simulations would vary by as much as 10 ppb on a seasonal mean basis since chemical destruction of boundary conditions in the base emissions scenario used in this study likely acts to reduce the degree to which ozone differences at the boundaries can influence surface ozone simulated within the

regional-scale CMAQ domain. The effect of this chemical destruction of boundary ozone on estimated boundary contributions to surface ozone under a base emissions scenario has been quantified by Baker et al. (2015). Such an expected range of up to 10 ppb in CMAQ-estimated seasonal mean NAB or USB values resulting from the use of boundary conditions derived from the four different large-scale models used in the present study would be consistent with the differences in NAB estimates reported by Fiore et al. (2014) that were derived from GEOS-Chem and AM3 applied for 2006.

**Code availability**: Source code for version 5.0.2 of the Community Multiscale Air Quality (CMAQ) modeling system can be downloaded from https://github.com/USEPA/CMAQ/tree/5.0.2. For further information, please visit the U.S. Environmental Protection Agency website for the CMAQ system: https://www.epa.gov/cmaq.

**Data availability**: Data used to generate figures and tables shown in this article can be downloaded at https://edg.epa.gov/metadata/catalog/main/home.page. Raw model outputs are available on request from the corresponding author. Observational data sets used in the analyses presented in this manuscript are available from their respective websites: https://aqs.epa.gov/aqsweb/airdata/download_files.html (AQS); https://www.epa.gov/castnet (CASTNET); http://www.woudc.org (WOUDC); https://www.esrl.noaa.gov/gmd/ozwv/ozsondes/ (NOAA ESRL).

**Competing interests**: The authors declare that they have no conflict of interest.

**Disclaimer:** The views expressed in this paper are those of the authors and do not necessarily represent the view or policies of the U.S. Environmental Protection Agency.

**Acknowledgements:** We gratefully acknowledge the Air Quality Model Evaluation International Initiative (AQMEII) and the Task Force on Hemispheric Transport of Air Pollution (TF-HTAP) for facilitating the analysis described in the manuscript by designing and coordinating internally consistent global- and regional-scale air quality model simulations. R.J. Park was supported by the National Strategic Project-Fine particle of the National Research Foundation of Korea(NRF) funded by the Ministry of Science and ICT(MSIT), the Ministry of Environment(ME), and the Ministry of Health and Welfare(MOHW) (NRF-2017M3D8A1090670). We also thank the maintainers of the AQS and CASTNET data portals from which the surface ozone observations used in this study were obtained. Finally, we would like to acknowledge NOAA ESRL for making available the ozonesonde measurements at Trinidad Head, Boulder, and Huntsville and the World Ozone and UV Data Center for making available the ozonesonde measurements at Edmonton, Churchill and Wallops Island.

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

**List of Figures**

Figure 1. Depiction of the vertical levels used in the four different large-scale models and the regional CMAQ model analyzed in this study. The pressure values were extracted for a location near the southwestern corner of the 12km CMAQ modeling domain and represent annual average values for 2010 at the midpoint of each vertical level. The dashed lines delineate the three pressure ranges (surface – 750 mb, 750 mb – 250 mb, and 250 mb – 50 mb) used for vertical integration in subsequent analyses.

Figure 2. Map of the 12 km CMAQ modeling domain, the five analysis domains, and the location of the AQS and CASTNET surface $O_3$ monitoring stations and ozonesonde launch sites.

Figure 3. Differences between observed and BASE modeled MDA8 ozone at AQS stations for each season and analysis region. For each season and region, the observed MDA8 ozone concentrations were rank ordered at each station. Next, differences between CMAQ simulations and observations were computed for each observed percentile either by selecting the model value corresponding to the date of the observed percentile (paired-in-time comparison, right column) or rank-ordering the model values and then selecting the modeled percentile corresponding to the observed percentile (unpaired-in-time comparison, left column). Finally, the median value of these paired-in-time and unpaired-in-time differences across all AQS stations in a given season and region was then computed for each observed percentile and is depicted in this figure

Figure 4. Profiles of BASE seasonal total ozone column mass changes $\Delta O_3$ for each CMAQ model layer due to the effects of horizontal advection (HADV), vertical advection (ZADV), horizontal diffusion (HDIF), vertical diffusion (VDIF), dry deposition (DDEP), chemistry (CHEM), and cloud processes including vertical mixing by convective clouds (CLDS). The values are summed over the entire modeling domain and represent the net change in ozone mass due to a given process in a given model layer and season.

Figure 5. Time series of monthly domain-wide total Process Analysis contributions to BASE ozone columns in CMAQ layers 1-21 (surface to approximately 750 mb), 22 – 31 (approximately 750 – 250 mb), and 32 – 35 (approximately 250 mb – 50 mb). The horizontal and vertical advection and diffusion terms were summed to compute the effects of total advection (TADV) and total diffusion (TDIF), respectively.

Figure 6. The upper three panels present time series of the monthly average domain total ozone column mass for the BASE, BC ZERO, EM ZERO, and NO O3 DDEP sensitivity simulations for the same three layer ranges analyzed in Figure 5 while the lowest panel presents time series of monthly average domain average ozone mixing ratios for the first model layer. The

dashed lines represent the 5th and 95th percentiles of the hourly domain total ozone column mass and domain average ozone mixing ratios for a given month.

Figure 7. Maps of differences in seasonal mean ozone mixing ratios between the three sensitivity simulations (BC ZERO, EM ZERO, and NO O3 DDEP) and the BASE simulation.

Figure 8a.Time-height cross sections of monthly mean ozone mixing ratios for ozonesonde observations (column 1), large-scale models (columns 2 – 5 in rows 1, 3, and 5), and regional CMAQ simulations (columns 2 – 5 in rows 2 and 4) at Trinidad Head, Edmonton, and Churchill. Note that no regional CMAQ results are shown for Churchill because the station is located outside the regional model domain. Additional details on the processing of observations and model simulations are provided in the text.

Figure 8b.Time-height cross sections of monthly mean ozone mixing ratios for ozonesonde observations (column 1), large-scale models (columns 2 – 5 in rows 1, 3, and 5), and regional CMAQ simulations (columns 2 – 5 in rows 2, 4, and 6) at Boulder, Huntsville, and Wallops Island. Additional details on the processing of observations and model simulations are provided in the text.

Figure 9: Monthly average time series of 500 mb observed ozone, ozone simulated by large-scale models (solid lines), and ozone simulated by regional CMAQ driven with boundary conditions from different large scale models (dashed lines) at Trinidad Head, Edmonton, Boulder, Huntsville, and Wallops Island. Additional details on the processing of observations and model simulations are provided in the text.

Figure 10. Maps of seasonal mean ozone mixing ratios at the surface for the BASE, BC H-CMAQ, BC GEOS-Chem, and BC-AM3 simulations. The left column shows the mixing ratios for the BASE simulation while the second, third and fourth columns show the differences between BC H-CMAQ and BASE, BC GEOS-Chem and BASE, and BC AM3 and BASE, respectively

Figure 11. Time series of differences between modeled and observed ozone mixing ratios. a) monthly means of daytime average mixing ratios at CASTNET monitors for regional model simulations, b) as in a) but for large-scale models, c) as in a) but for AQS monitors, d) as in c) but for monthly means of MDA8 instead of monthly means of daytime average mixing ratios, e) as in c) but for daily daytime average mixing ratios, and f) as in d) but for daily MDA8.

Figure 12. Map of seasonal mean bias for daytime average ozone at CASTNET sites for BASE, BC H-CMAQ, BC GEOS-Chem, and BC AM3 (rows 1 and 3) and C-IFS, H-CMAQ, GEOS-Chem, and AM3 (rows 2 and 4) for spring (rows 1 and 2) and summer (rows 3 and 4)

Figure 13. Paired-in-time differences between observed and modeled MDA8 ozone at AQS stations for each season and analysis region. Model results are for BASE (red), BC H-CMAQ (blue), BC GEOS-Chem (green), and BC AM3 (orange). For each season and region, the observed MDA8 ozone concentrations were rank ordered at each station. Next, differences between CMAQ simulations and observations were computed for each observed percentile by selecting the model value corresponding to the date of the observed percentile. Finally, the median value of these paired-in-time differences across all AQS stations in a given season and region was then computed for each observed percentile and is depicted in this figure.

**Table 1. Mapping of gas-phase species from C-IFS, H-CMAQ, GEOS-Chem and AM3 to regional-scale CMAQ**

| CMAQv5.0.2 CB05-TUCL Target Species | C-IFS Species | H-CMAQ Species | GEOS-Chem Species | AM3 Species |
|---|---|---|---|---|
| $O_3$ | $O_3$ | $O_3$ | $O_x$-$NO_x$ | $O_3$ |
| CO | CO | CO | CO | CO |
| FORM | $CH_2O$ | FORM | $CH_2O$ | |
| NO | NO | NO | NO | NO |
| $NO_2$ | $NO_2$ | $NO_2$ | $NO_2$ | $NO_2$ |
| $HNO_3$ | $HNO_3$ | $HNO_3$ | $HNO_3$ | $HNO_3$ |
| $N_2O_5$ | | $N_2O_5$ | $N_2O_5$ | |
| PAN | PAN | PAN | PAN | PAN |
| PANX | | PANX | PPN, PMN | |
| $SO_2$ | $SO_2$ | $SO_2$ | $SO_2$ | $SO_2$ |
| PAR | PAR, $CH_3COCH_3$, $C_3H_8$ | PAR | $C_3H_8$, ALK4, ACET, MEK, BENZ | ACETONE, PROPANE |
| ETHA | $C_2H_6$ | ETHA | $C_2H_6$ | $C_2H_6$ |
| MEOH | $CH_3OH$ | MEOH | | |
| ETOH | $C_2H_5OH$ | ETOH | | |
| ETH | $C_2H_4$ | ETH | | |
| ALD2 | ALD2 | ALD2 | ALD2 | |
| OLE | OLE | OLE | PRPE | |
| ISOP | ISOP | ISOP | ISOP | |
| ISPD | | ISPD | MACR, MVK | |
| FACD | HCOOH | FACD | | |
| MEPX | CH3OOH | MEPX | MP | |
| NTR | ONIT | NTROH, NTRALK, NTRCN, NTRCNOH, NTRM, NTRI, NTRPX | R4N2 | |
| PNA | | PNA | $HNO_4$ | |
| $H_2O_2$ | | $H_2O_2$ | $H_2O_2$ | |
| IOLE | | IOLE | PRPE | |
| TOL | | TOL | TOLU | |
| XYL | | XYL | XYLE | |
| BENZENE | | BENZENE | BENZ | |

**Table 2. List of regional-scale CMAQ simulations**

| Acronym | Lateral Boundary Conditions | Emissions | CMAQ configuration |
|---|---|---|---|
| BASE | C-IFS | Pouliot et al. (2015) | Solazzo et al. (2017a) Hogrefe et al. (2017) |
| BC ZERO | Zero for all species | Pouliot et al. (2015) | Solazzo et al. (2017a) Hogrefe et al. (2017) |
| EM ZERO | C-IFS | Zero for anthropogenic and wildfire emissions within the CMAQ modeling domain | Solazzo et al. (2017a) Hogrefe et al. (2017) |
| NO O3 DDEP | C-IFS | Pouliot et al. (2015) | Solazzo et al. (2017a) Hogrefe et al. (2017) Modified to "turn off" ozone dry deposition |
| BC H-CMAQ | H-CMAQ | Pouliot et al. (2015) | Solazzo et al. (2017a) Hogrefe et al. (2017) |
| BC GEOS-Chem | GEOS-Chem | Pouliot et al. (2015) | Solazzo et al. (2017a) Hogrefe et al. (2017) |
| BC AM3 | AM3 | Pouliot et al. (2015) | Solazzo et al. (2017a) Hogrefe et al. (2017) |

**Table 3. Seasonal model performance for MDA8 ozone at AQS sites as measured by Normalized Mean Bias (NMB), Normalized Mean Error (NME), and correlation coefficient for the BASE simulations for all sites and the five analysis regions shown in Figure 2. The metrics were computed at each AQS site for each season and the median metric across all sites in a given region and season is shown in in this table. The cells in the table are color-coded based on the model evaluation goals and acceptability criteria proposed by Emery et al. (2017). Green cells indicate regions and seasons where model performance meets the goal for a given metric (NMB <+- 5%, NME <15%, and R > 0.75), yellow cells indicate regions and seasons where model performance meets the acceptability criterion but not the goal (+-5% < NMB <+- 15%, 15% < NME <25%, and 0.5 < R < 0.75), and orange cells indicate regions and seasons where neither the goal nor the acceptability criterion are met (NMB >= +/- 15%, NME >= 25%, R <= 0.5)**

|       |        | All   | NW    | IMW   | MW    | SE    | NE    |
|-------|--------|-------|-------|-------|-------|-------|-------|
| NMB   | Spring | 2.1   | 9.1   | 3.6   | 2.0   | 2.6   | -3.2  |
|       | Summer | 6.8   | 13.5  | -3.2  | 6.3   | 9.6   | 2.6   |
|       | Fall   | 3.1   | 13.0  | 2.1   | 1.0   | 2.1   | 1.7   |
|       | Winter | -5.1  | 11.0  | -2.8  | -23.1 | -7.3  | -28.2 |
| NME   | Spring | 11.3  | 13.1  | 8.5   | 11.3  | 11.3  | 12.0  |
|       | Summer | 13.6  | 16.2  | 10.5  | 13.3  | 14.7  | 12.0  |
|       | Fall   | 13.5  | 20.8  | 9.4   | 12.1  | 11.7  | 14.4  |
|       | Winter | 18.8  | 20.5  | 19.2  | 28.8  | 12.0  | 29.1  |
| R     | Spring | 0.75  | 0.57  | 0.66  | 0.8   | 0.82  | 0.74  |
|       | Summer | 0.73  | 0.78  | 0.67  | 0.7   | 0.72  | 0.82  |
|       | Fall   | 0.82  | 0.72  | 0.74  | 0.84  | 0.81  | 0.87  |
|       | Winter | 0.65  | 0.57  | 0.6   | 0.75  | 0.76  | 0.69  |

**Table 4a. NMB for MDA8 ozone at AQS sites for BASE, BC H-CMAQ, BC GEOS-Chem, and BC AM3. See the legend for Table 3 for a description on how the metrics were computed and for a definition of the color coding used in this Table.**

|  |  | All | NW | IMW | MW | SE | NE |
|---|---|---|---|---|---|---|---|
| Spring | BASE | 2.1 | 9.1 | 3.6 | 2.0 | 2.6 | -3.2 |
|  | BC H-CMAQ | 5.3 | 10.3 | 5.2 | 5.0 | 5.8 | 0.7 |
|  | BC GEOS-Chem | 2.1 | 8.9 | 1.0 | 2.5 | 3.3 | -1.7 |
|  | BC AM3 | 18.6 | 31.6 | 23.4 | 18.2 | 14.8 | 16.2 |
| Summer | BASE | 6.8 | 13.5 | -3.2 | 6.3 | 9.6 | 2.6 |
|  | BC H-CMAQ | 8.8 | 11.8 | 2.0 | 7.5 | 12.4 | 3.2 |
|  | BC GEOS-Chem | 7.5 | 8.9 | -1.2 | 6.7 | 11.1 | 2.8 |
|  | BC AM3 | 11.4 | 23.0 | 2.9 | 10.6 | 12.1 | 8.4 |
| Fall | BASE | 3.1 | 13.0 | 2.1 | 1.0 | 2.1 | 1.7 |
|  | BC H-CMAQ | 9.2 | 18.4 | 8.7 | 7.1 | 7.7 | 8.8 |
|  | BC GEOS-Chem | 6.3 | 15.9 | 5.0 | 5.0 | 4.7 | 6.3 |
|  | BC AM3 | 13.3 | 30.4 | 12.3 | 13.7 | 10.7 | 15.1 |
| Winter | BASE | -5.1 | 11.0 | -2.8 | -23.1 | -7.3 | -28.2 |
|  | BC H-CMAQ | 11.3 | 27.2 | 9.2 | 3.5 | 12.1 | 1.3 |
|  | BC GEOS-Chem | 2.2 | 16.6 | 3.3 | -8.3 | 1.2 | -12.9 |
|  | BC AM3 | 18.4 | 36.4 | 19.6 | 13.5 | 17.9 | 12.1 |

**Table 4b. NME for MDA8 ozone at AQS sites for BASE, BC H-CMAQ, BC GEOS-Chem, and BC AM3. See the legend for Table 3 for a description on how the metrics were computed and for a definition of the color coding used in this Table.**

| | | All | NW | IMW | MW | SE | NE |
|---|---|---|---|---|---|---|---|
| Spring | BASE | 11.3 | 13.1 | 8.5 | 11.3 | 11.3 | 12.0 |
| | BC H-CMAQ | 12.1 | 13.3 | 10.6 | 11.9 | 11.7 | 11.6 |
| | BC GEOS-Chem | 11.1 | 13.6 | 8.6 | 11.2 | 11.1 | 11.2 |
| | BC AM3 | 20.2 | 31.6 | 23.7 | 20.1 | 17.1 | 18.9 |
| Summer | BASE | 13.6 | 16.2 | 10.5 | 13.3 | 14.7 | 12.0 |
| | BC H-CMAQ | 14.9 | 15.4 | 11.4 | 14.6 | 16.3 | 13.2 |
| | BC GEOS-Chem | 14.0 | 13.9 | 10.8 | 13.6 | 15.4 | 12.3 |
| | BC AM3 | 16.0 | 24.7 | 13.8 | 15.2 | 16.1 | 13.8 |
| Fall | BASE | 13.5 | 20.8 | 9.4 | 12.1 | 11.7 | 14.4 |
| | BC H-CMAQ | 15.2 | 22.0 | 12.2 | 14.2 | 13.2 | 16.1 |
| | BC GEOS-Chem | 14.3 | 20.6 | 10.0 | 13.7 | 12.7 | 15.9 |
| | BC AM3 | 17.9 | 30.5 | 14.5 | 18.1 | 14.6 | 20.0 |
| Winter | BASE | 18.8 | 20.5 | 19.2 | 28.8 | 12.0 | 29.1 |
| | BC H-CMAQ | 17.4 | 30.5 | 18.0 | 15.9 | 15.1 | 14.4 |
| | BC GEOS-Chem | 15.8 | 21.3 | 17.0 | 19.0 | 11.0 | 16.8 |
| | BC AM3 | 21.2 | 38.7 | 22.7 | 18.0 | 19.4 | 15.5 |

**Table 4c. R for MDA8 ozone at AQS sites for BASE, BC H-CMAQ, BC GEOS-Chem, and BC AM3. See the legend for Table 3 for a description on how the metrics were computed and for a definition of the color coding used in this Table.**

| | | All | NW | IMW | MW | SE | NE |
|---|---|---|---|---|---|---|---|
| Spring | BASE | 0.75 | 0.57 | 0.66 | 0.8 | 0.82 | 0.74 |
| | BC H-CMAQ | 0.72 | 0.59 | 0.55 | 0.8 | 0.82 | 0.72 |
| | BC GEOS-Chem | 0.74 | 0.57 | 0.62 | 0.81 | 0.83 | 0.75 |
| | BC AM3 | 0.63 | 0.45 | 0.6 | 0.65 | 0.77 | 0.59 |
| Summer | BASE | 0.73 | 0.78 | 0.67 | 0.7 | 0.72 | 0.82 |
| | BC H-CMAQ | 0.72 | 0.78 | 0.67 | 0.68 | 0.71 | 0.8 |
| | BC GEOS-Chem | 0.70 | 0.78 | 0.66 | 0.68 | 0.68 | 0.81 |
| | BC AM3 | 0.71 | 0.64 | 0.56 | 0.69 | 0.72 | 0.79 |
| Fall | BASE | 0.82 | 0.72 | 0.74 | 0.84 | 0.81 | 0.87 |
| | BC H-CMAQ | 0.80 | 0.71 | 0.72 | 0.81 | 0.81 | 0.86 |
| | BC GEOS-Chem | 0.80 | 0.71 | 0.74 | 0.81 | 0.79 | 0.86 |
| | BC AM3 | 0.77 | 0.67 | 0.59 | 0.75 | 0.8 | 0.83 |
| Winter | BASE | 0.65 | 0.57 | 0.6 | 0.75 | 0.76 | 0.69 |
| | BC H-CMAQ | 0.67 | 0.71 | 0.63 | 0.75 | 0.71 | 0.7 |
| | BC GEOS-Chem | 0.67 | 0.68 | 0.64 | 0.77 | 0.75 | 0.71 |
| | BC AM3 | 0.68 | 0.71 | 0.64 | 0.78 | 0.72 | 0.73 |

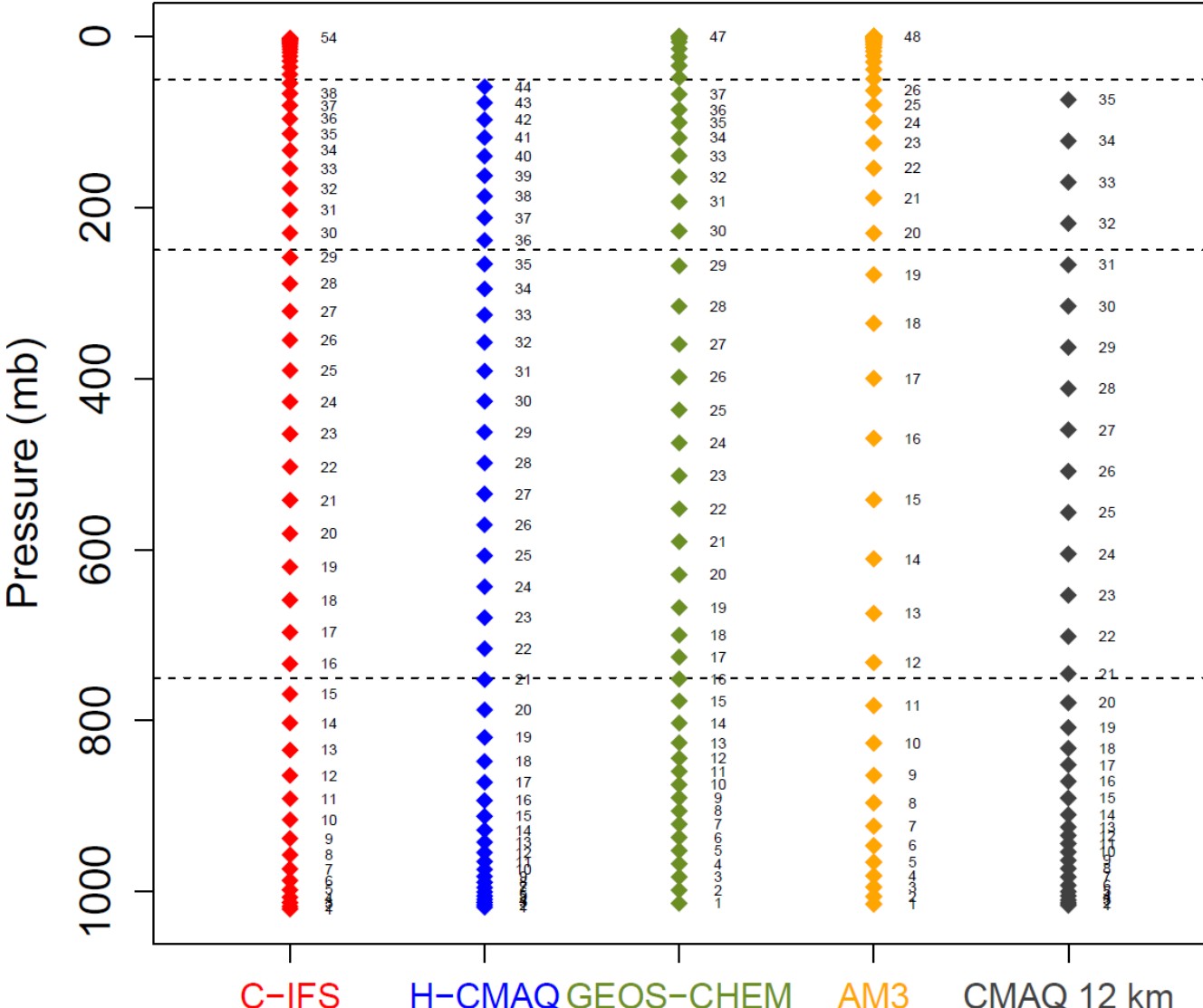

**Figure 1. Depiction of the vertical levels used in the four different large-scale models and the regional CMAQ model analyzed in this study. The pressure values were extracted for a location near the southwestern corner of the 12km CMAQ modeling domain and represent annual average values for 2010 at the midpoint of each vertical level. The dashed lines delineate the three pressure ranges (surface – 750 mb, 750 mb – 250 mb, and 250 mb – 50 mb) used for vertical integration in subsequent analyses.**

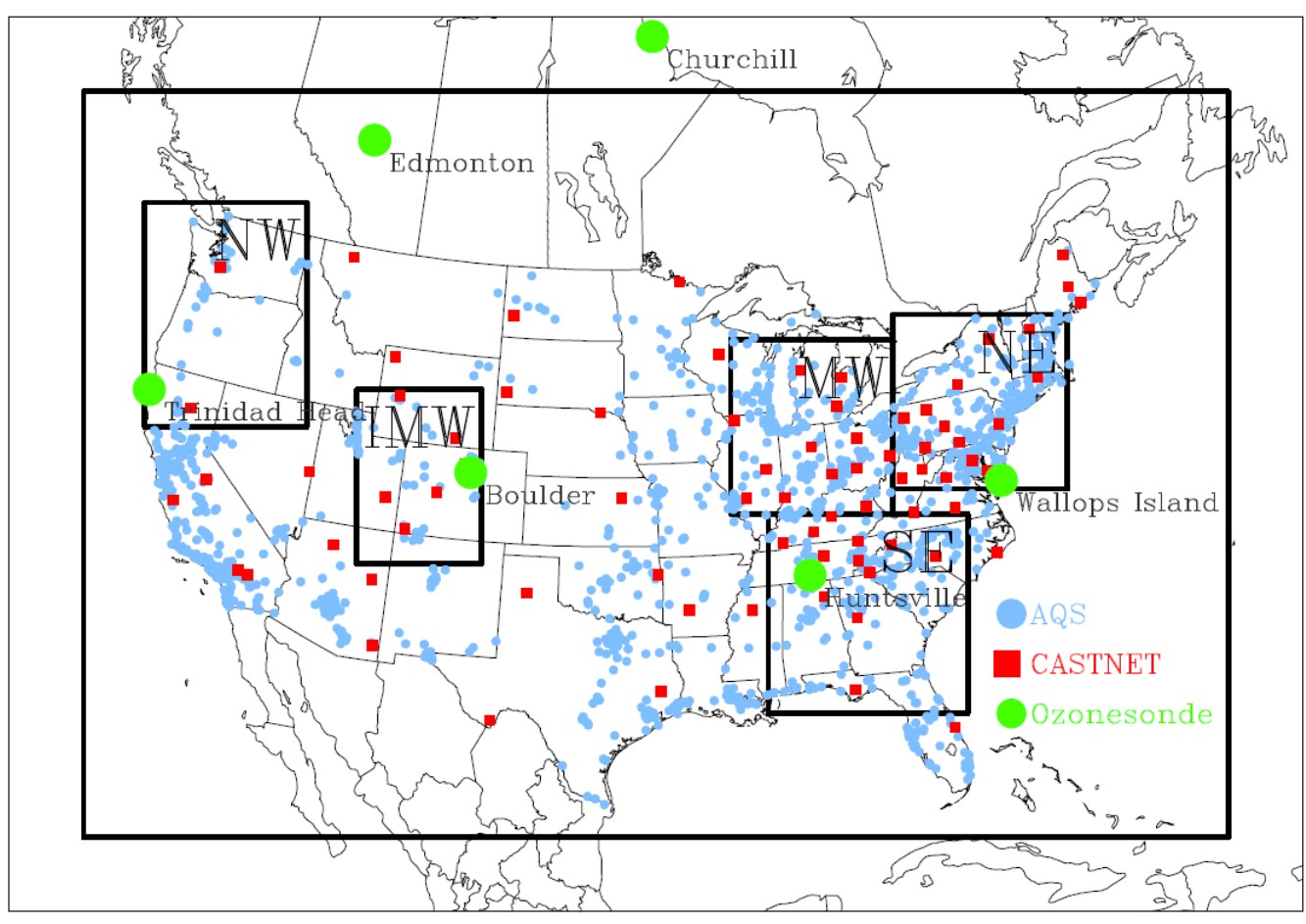

**Figure 2. Map of the 12 km CMAQ modeling domain, the five analysis domains, and the location of the AQS and CASTNET surface O₃ monitoring stations and ozonesonde launch sites.**

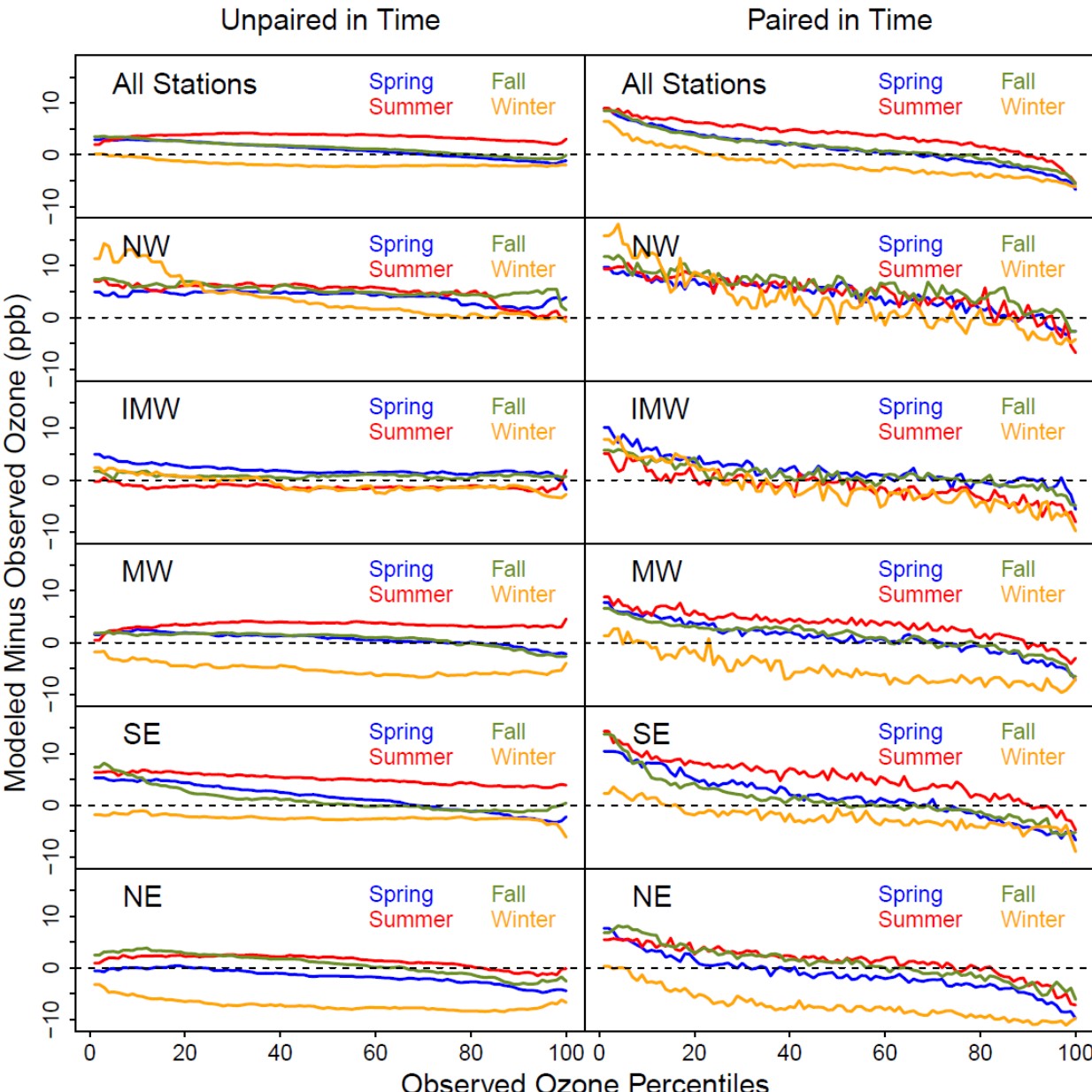

**Figure 3. Differences between observed and BASE modeled MDA8 ozone at AQS stations for each season and analysis region. For each season and region, the observed MDA8 ozone concentrations were rank ordered at each station. Next, differences between the BASE simulations and observations were computed for each observed percentile either by selecting the model value corresponding to the date of the observed percentile (paired-in-time comparison, right column) or rank-ordering the model values and then selecting the modeled percentile corresponding to the observed percentile (unpaired-in-time comparison, left column). Finally, the median value of these paired-in-time and unpaired-in-time differences across all AQS stations in a given season and region was then computed for each observed percentile and is depicted in this figure.**

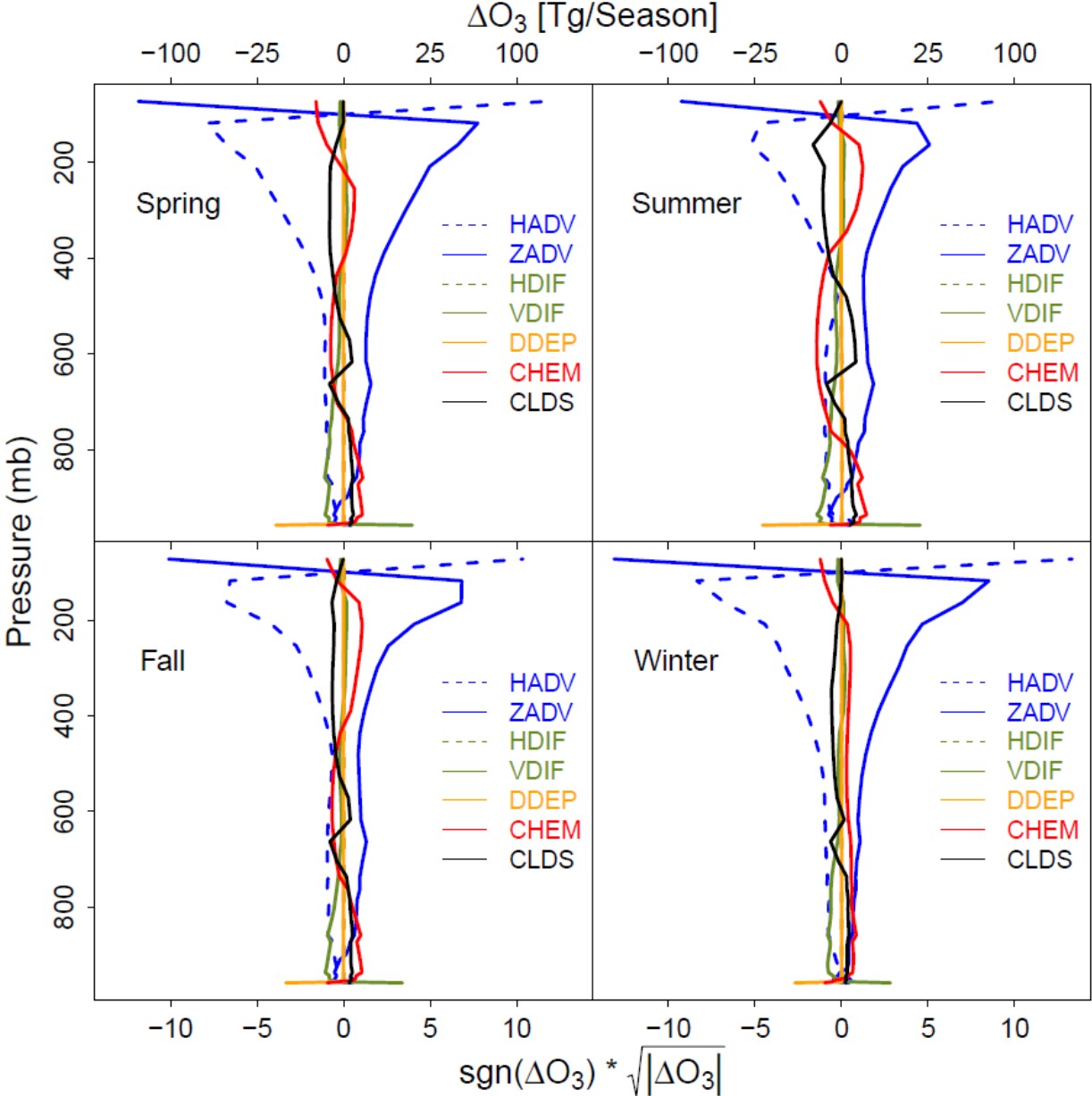

**Figure 4. Profiles of BASE seasonal total ozone column mass changes ΔO₃ for each CMAQ model layer due to the effects of horizontal advection (HADV), vertical advection (ZADV), horizontal diffusion (HDIF), vertical diffusion (VDIF), dry deposition (DDEP), chemistry (CHEM), and cloud processes including vertical mixing by convective clouds (CLDS). The values are summed over the entire modeling domain and represent the net change in ozone mass due to a given process in a given model layer and season.**

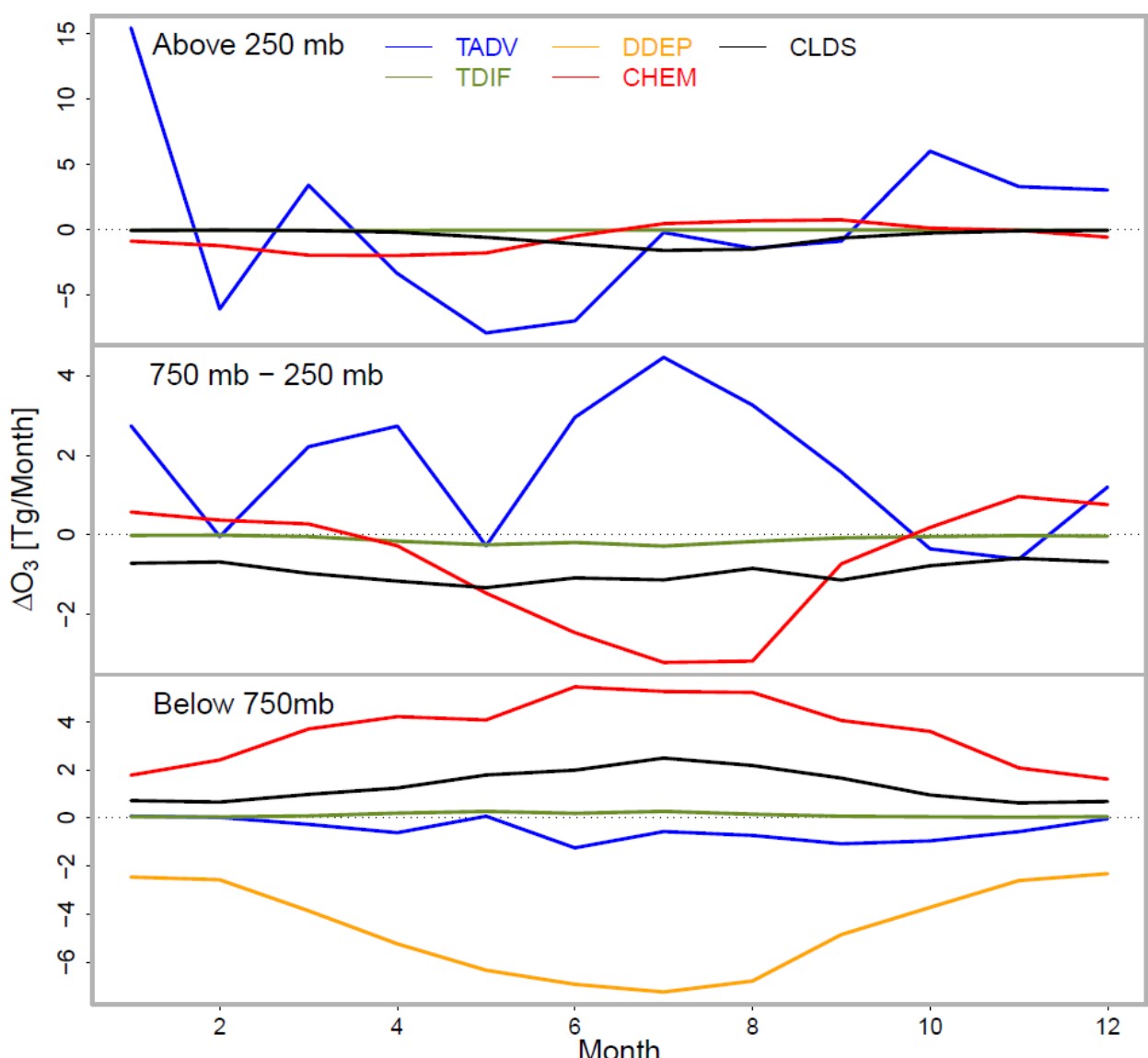

**Figure 5. Time series of monthly domain-wide total PA contributions to BASE ozone columns in CMAQ layers 1-21 (surface to approximately 750 mb), 22 – 31 (approximately 750 – 250 mb), and 32 – 35 (approximately 250 mb – 50 mb). The horizontal and vertical advection and diffusion terms were summed to compute the effects of total advection (TADV) and total diffusion (TDIF), respectively.**

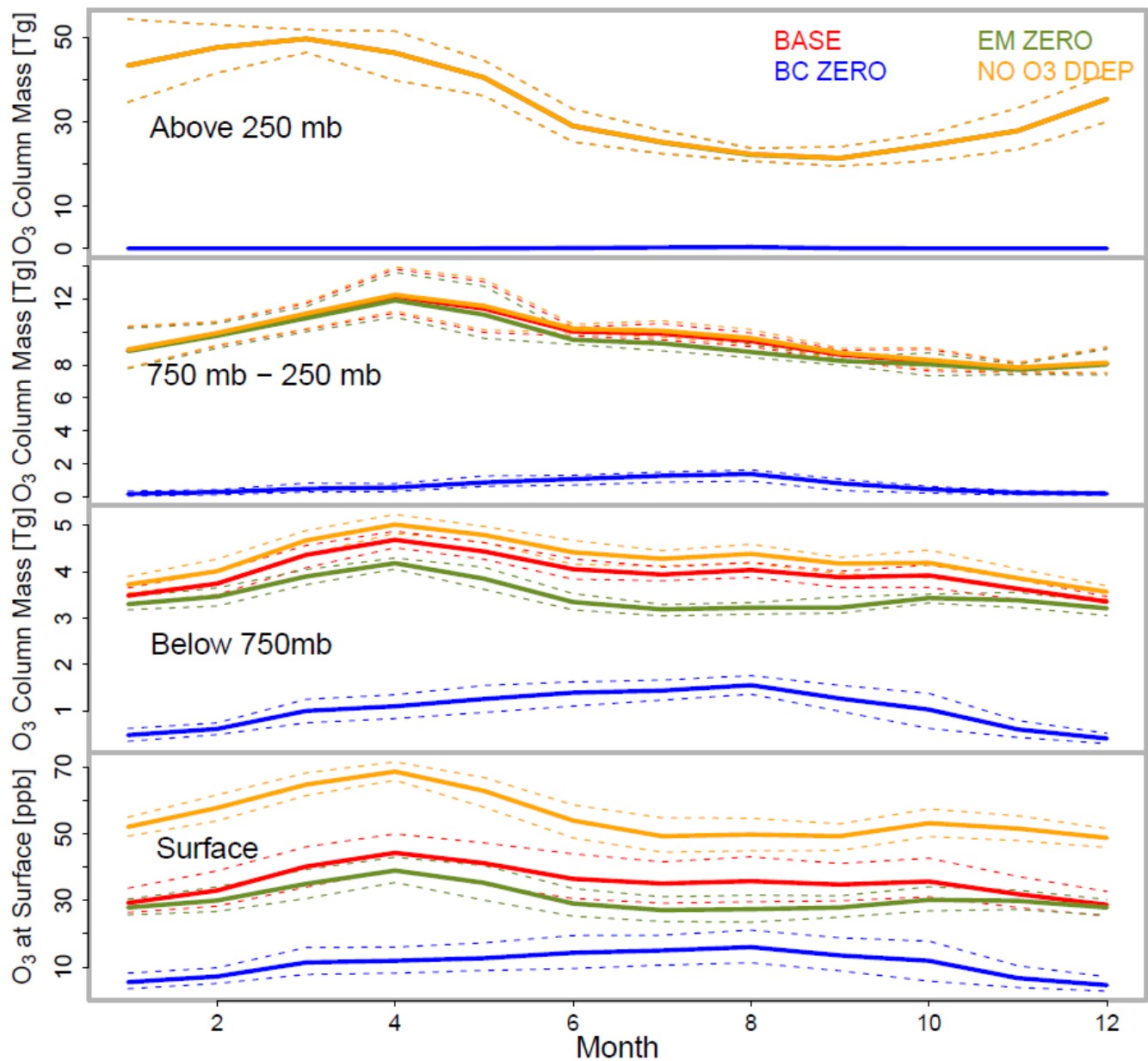

**Figure 6. The upper three panels present time series of the monthly average domain total ozone column mass for the BASE, BC ZERO, EM ZERO, and NO O3 DDEP sensitivity simulations for the same three layer ranges analyzed in Figure 5 while the lowest panel presents time series of monthly average domain average ozone mixing ratios for the first model layer. The dashed lines represent the 5th and 95th percentiles of the hourly domain total ozone column mass and domain average ozone mixing ratios for a given month.**

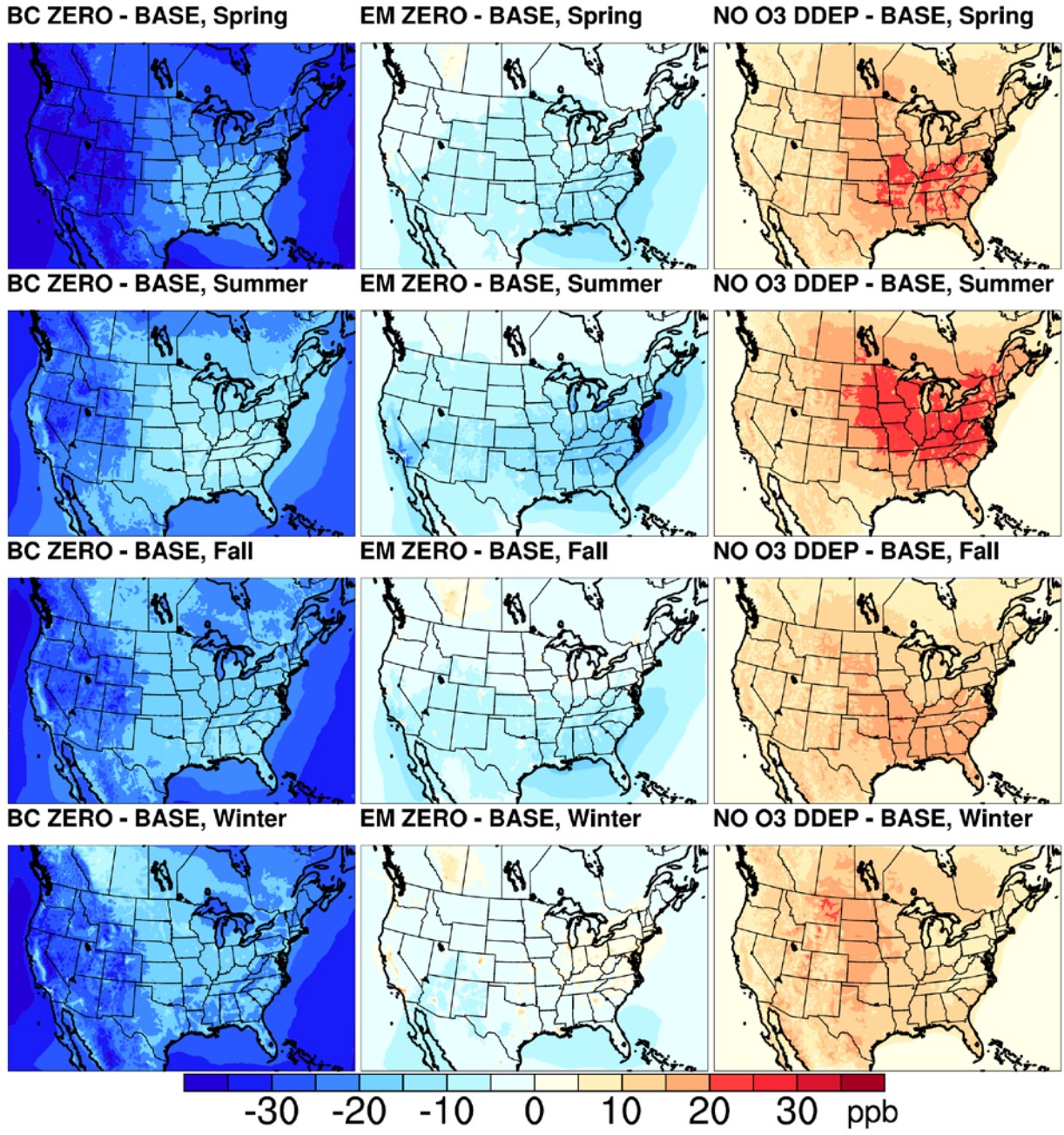

**Figure 7. Maps of differences in seasonal mean ozone mixing ratios between the three sensitivity simulations (BC ZERO, EM ZERO, and NO O3 DDEP) and the BASE simulation.**

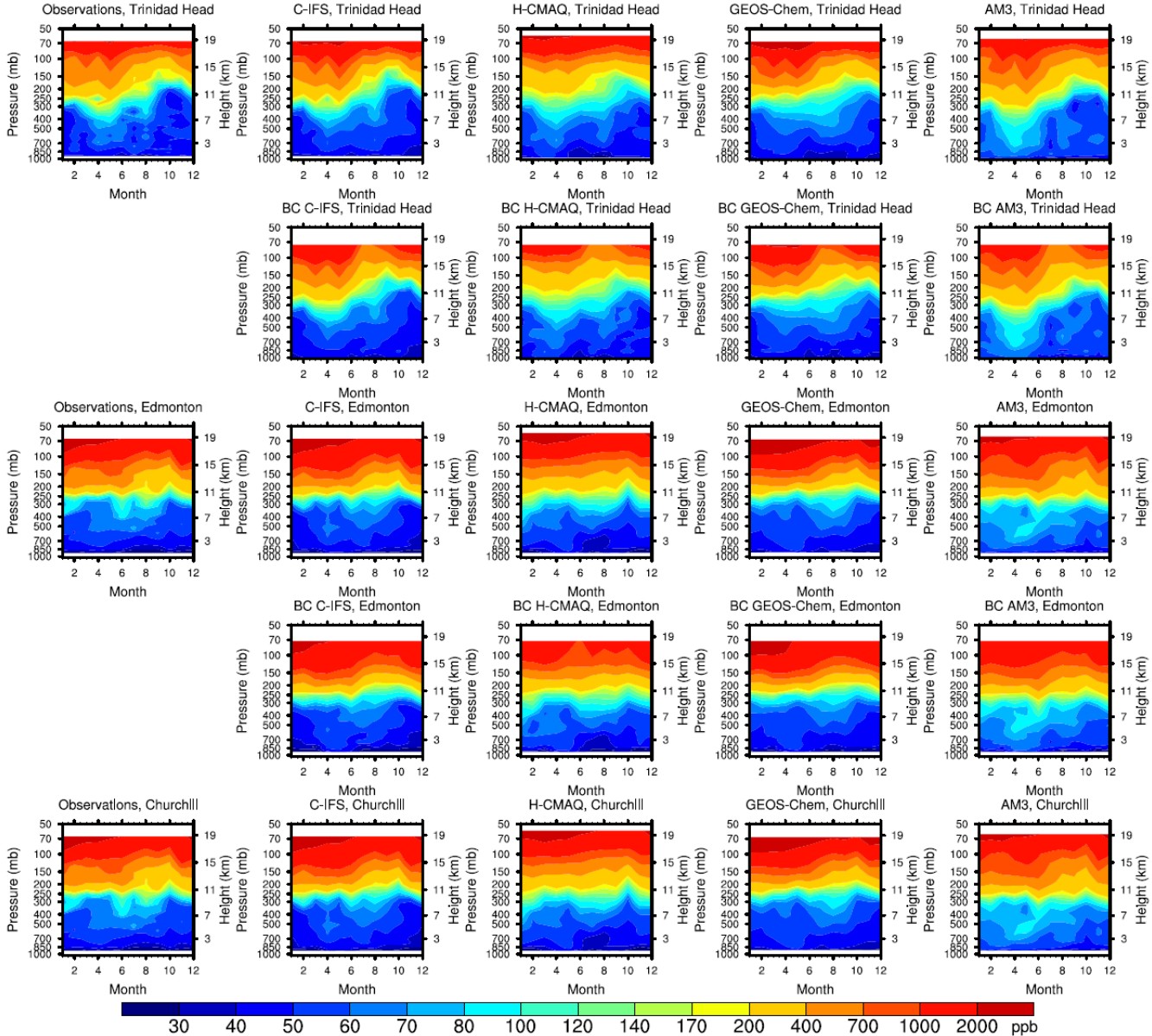

**Figure 8a. Time-height cross sections of monthly mean ozone mixing ratios for ozonesonde observations (column 1), large-scale models (columns 2 – 5 in rows 1, 3, and 5), and regional CMAQ simulations (columns 2 – 5 in rows 2 and 4) at Trinidad Head, Edmonton, and Churchill. Note that no regional CMAQ results are shown for Churchill because the station is located outside the regional model domain. Additional details on the processing of observations and model simulations are provided in the text.**

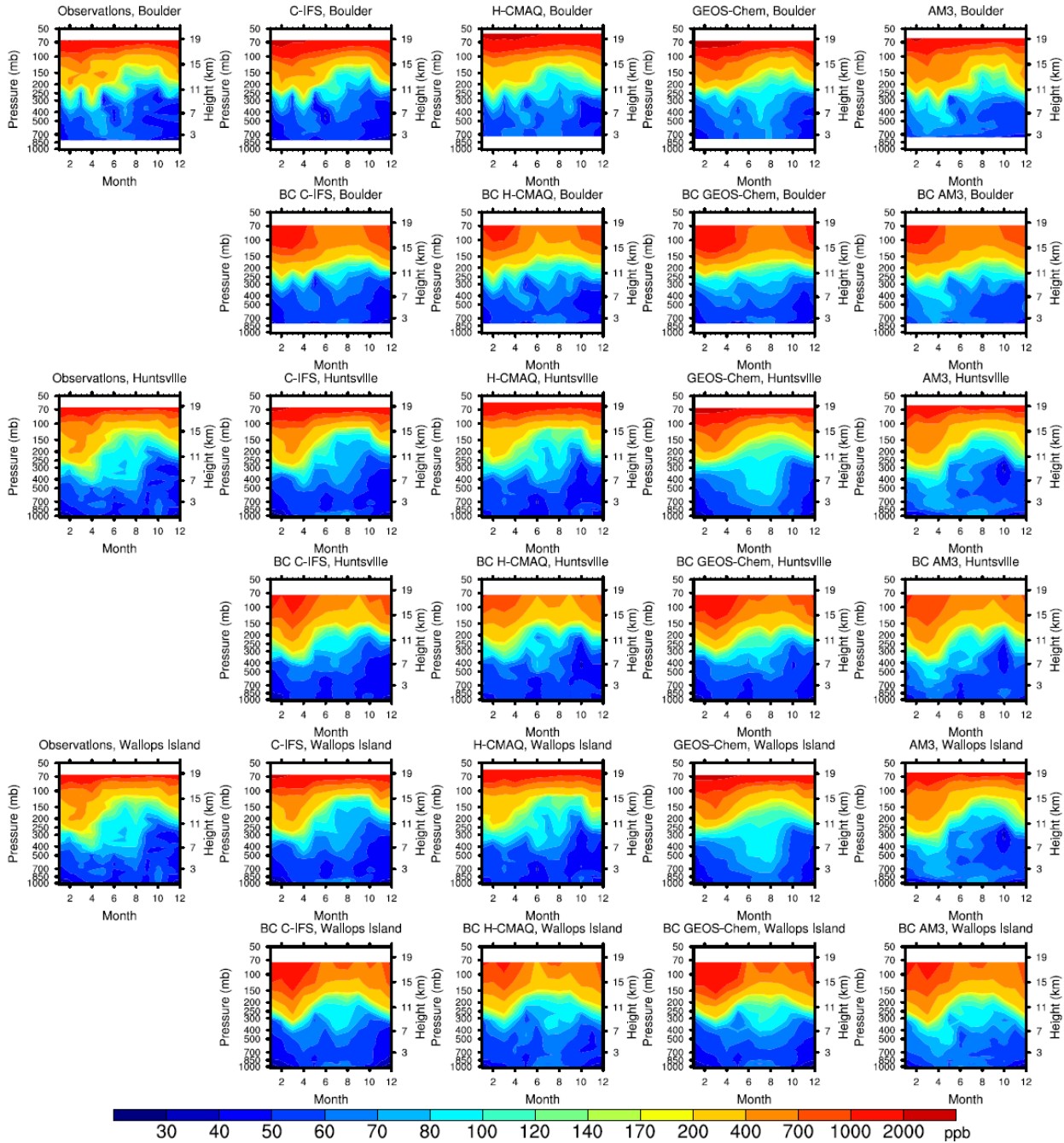

**Figure 8b. Time-height cross sections of monthly mean ozone mixing ratios for ozonesonde observations (column 1), large-scale models (columns 2 – 5 in rows 1, 3, and 5), and regional CMAQ simulations (columns 2 – 5 in rows 2, 4, and 6) at Boulder, Huntsville, and Wallops Island. Additional details on the processing of observations and model simulations are provided in the text.**

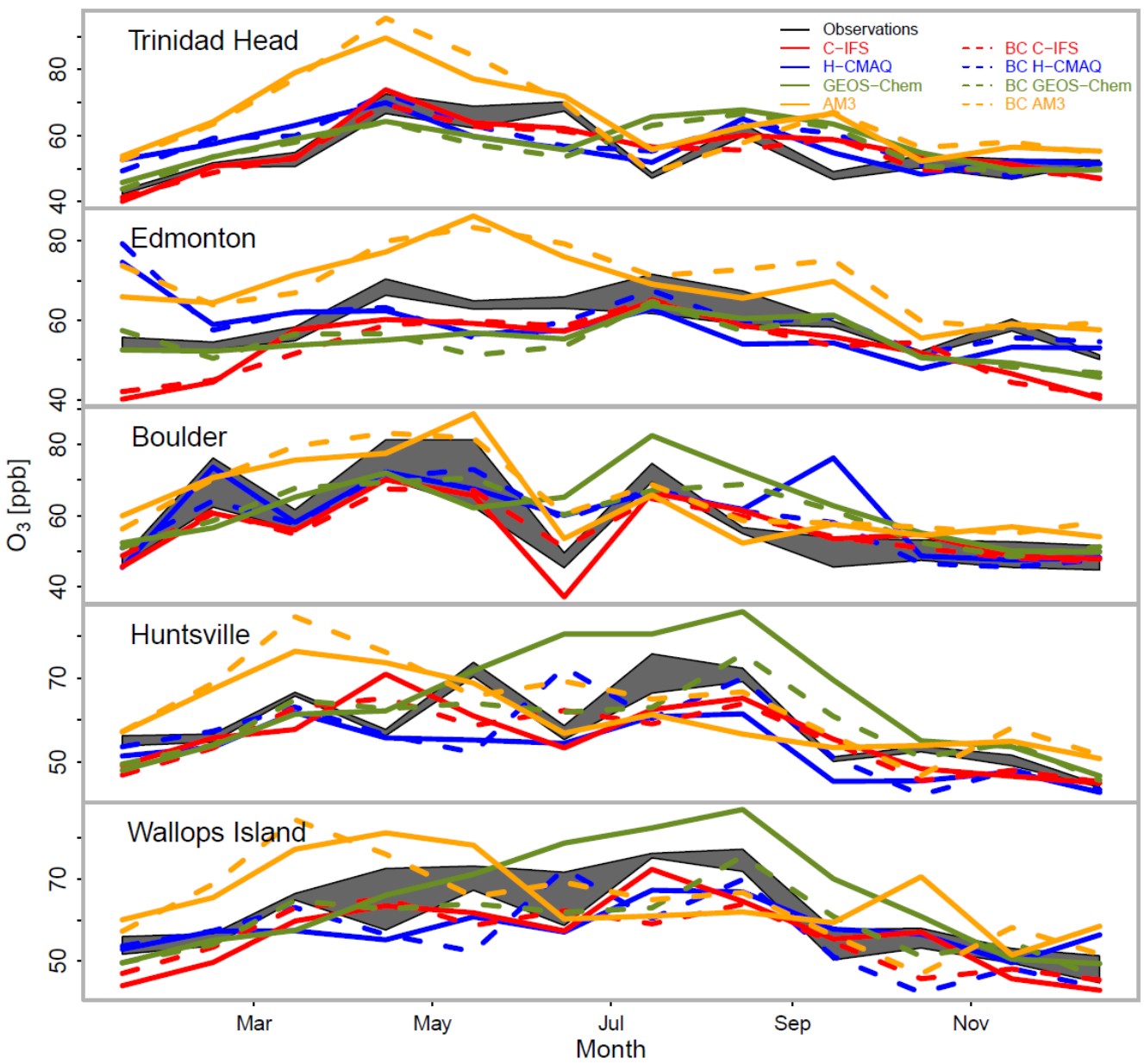

**Figure 9: Monthly average time series of 500 mb observed ozone, ozone simulated by large-scale models (solid lines), and ozone simulated by regional CMAQ driven with boundary conditions from different large scale models (dashed lines) at Trinidad Head, Edmonton, Boulder, Huntsville, and Wallops Island. Additional details on the processing of observations and model simulations are provided in the text.**

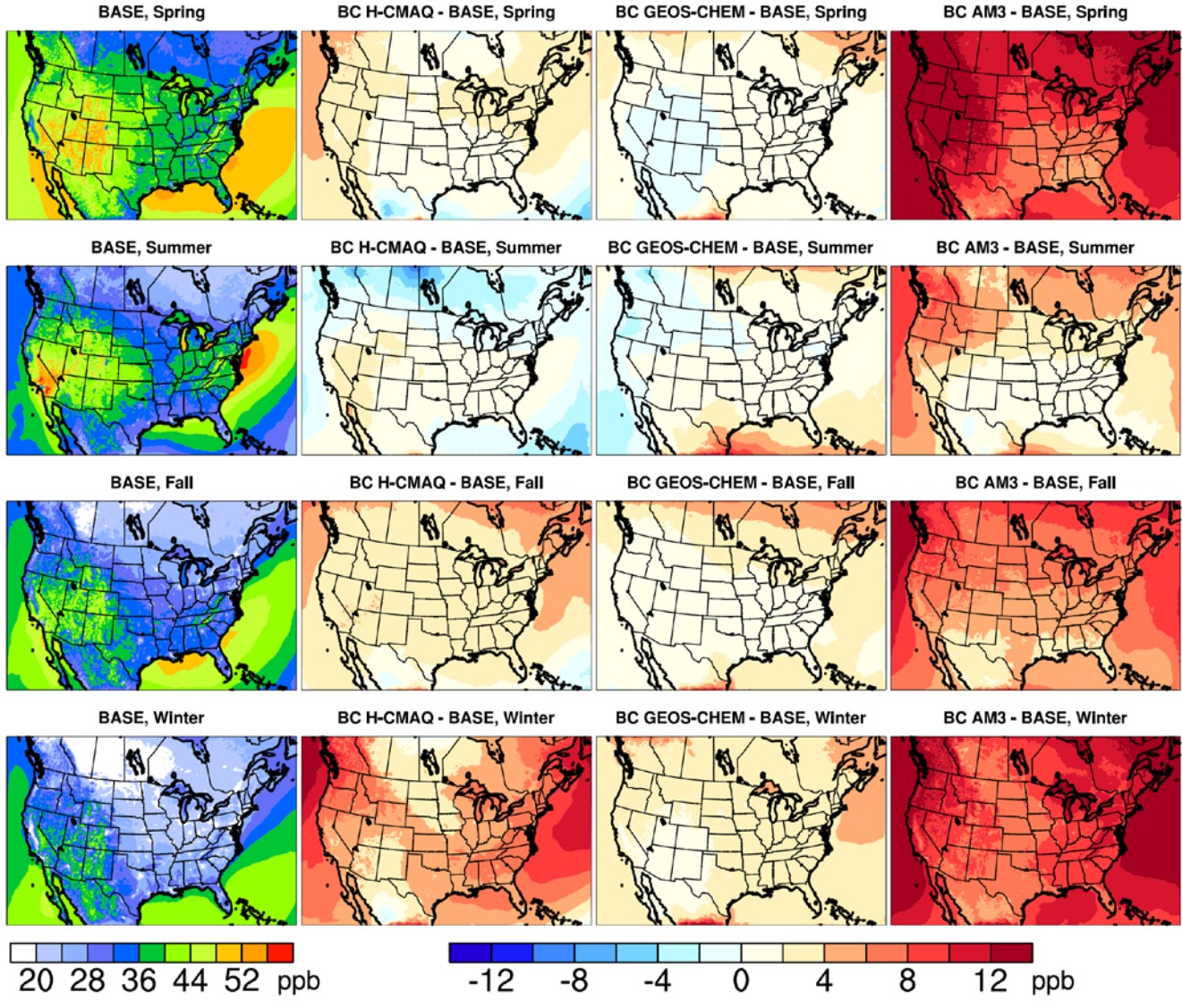

**Figure 10. Maps of seasonal mean ozone mixing ratios at the surface for the BASE, BC H-CMAQ, BC GEOS-Chem, and BC-AM3 simulations. The left column shows the mixing ratios for the BASE simulation while the second, third and fourth columns show the differences between BC H-CMAQ and BASE, BC GEOS-Chem and BASE, and BC AM3 and BASE, respectively**

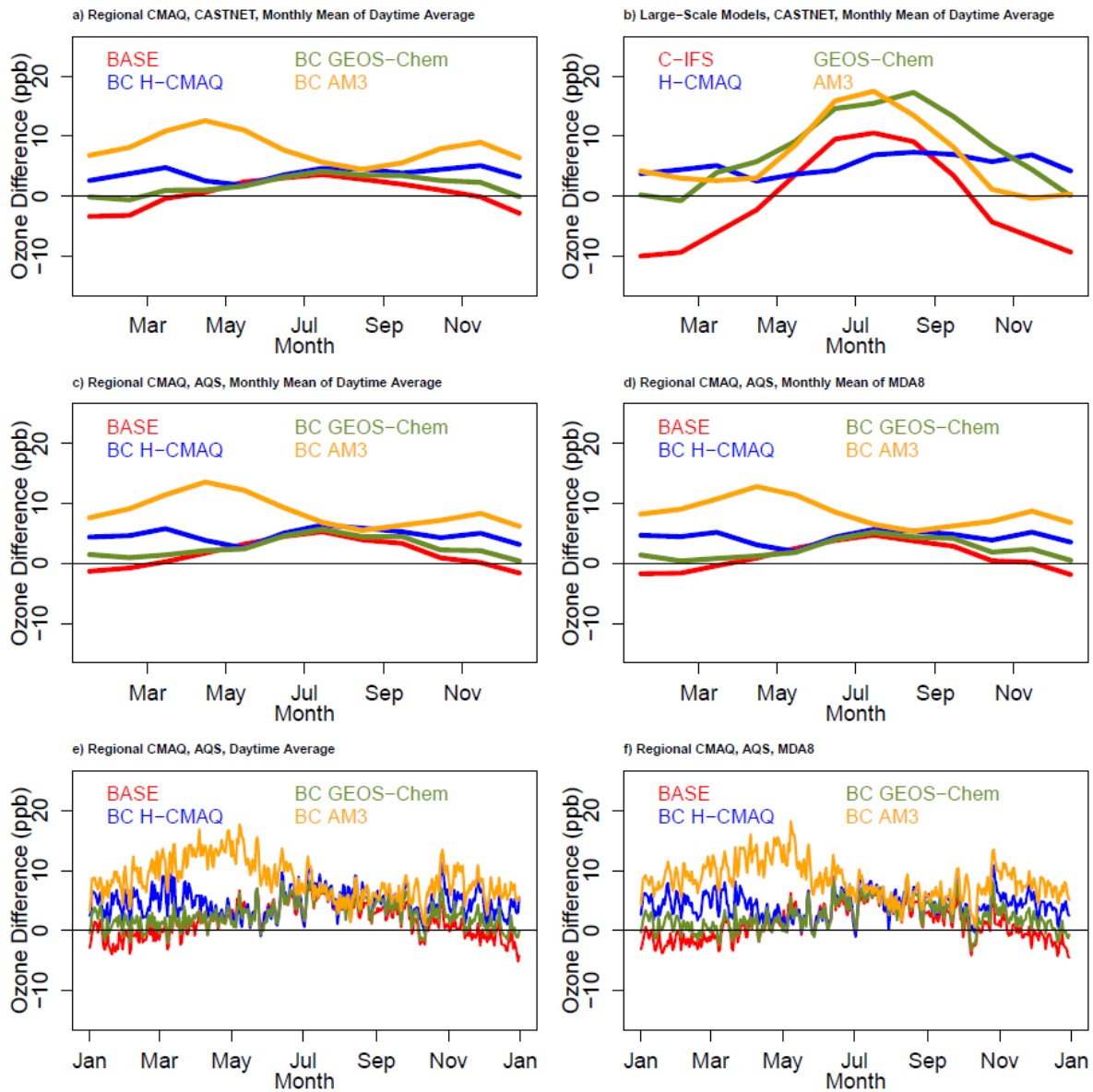

**Figure 11. Time series of differences between modeled and observed ozone mixing ratios. a) monthly means of daytime average mixing ratios at CASTNET monitors for regional model simulations, b) as in a) but for large-scale models, c) as in a) but for AQS monitors, d) as in c) but for monthly means of MDA8 instead of monthly means of daytime average mixing ratios, e) as in c) but for daily daytime average mixing ratios, and f) as in d) but for daily MDA8.**

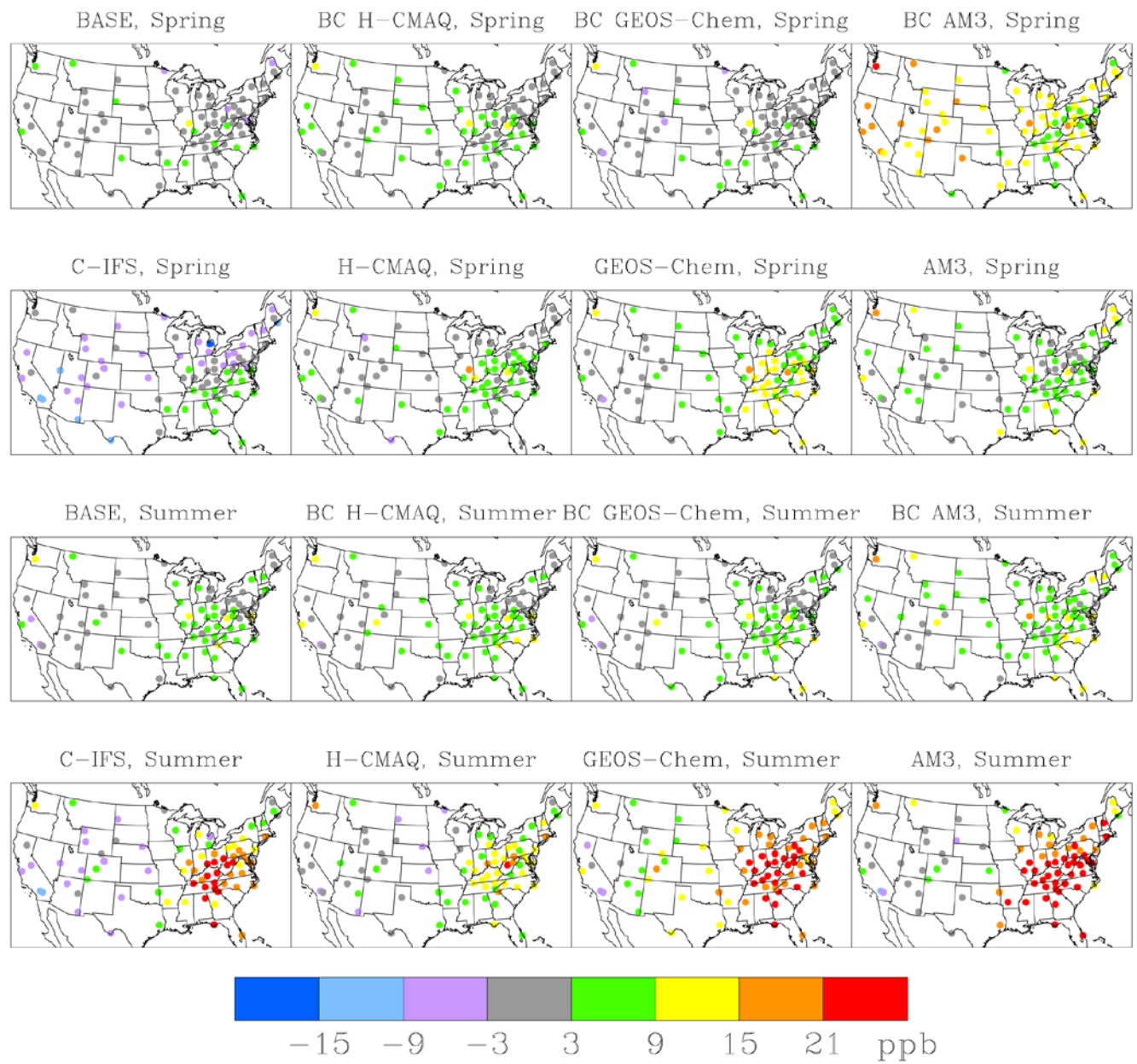

**Figure 12. Map of mean bias for daytime average ozone at CASTNET sites for BASE, BC H-CMAQ, BC GEOS-Chem, and BC AM3 (rows 1 and 3) and C-IFS, H-CMAQ, GEOS-Chem, and AM3 (rows 2 and 4) for spring (rows 1 and 2) and summer (rows 3 and 4)**

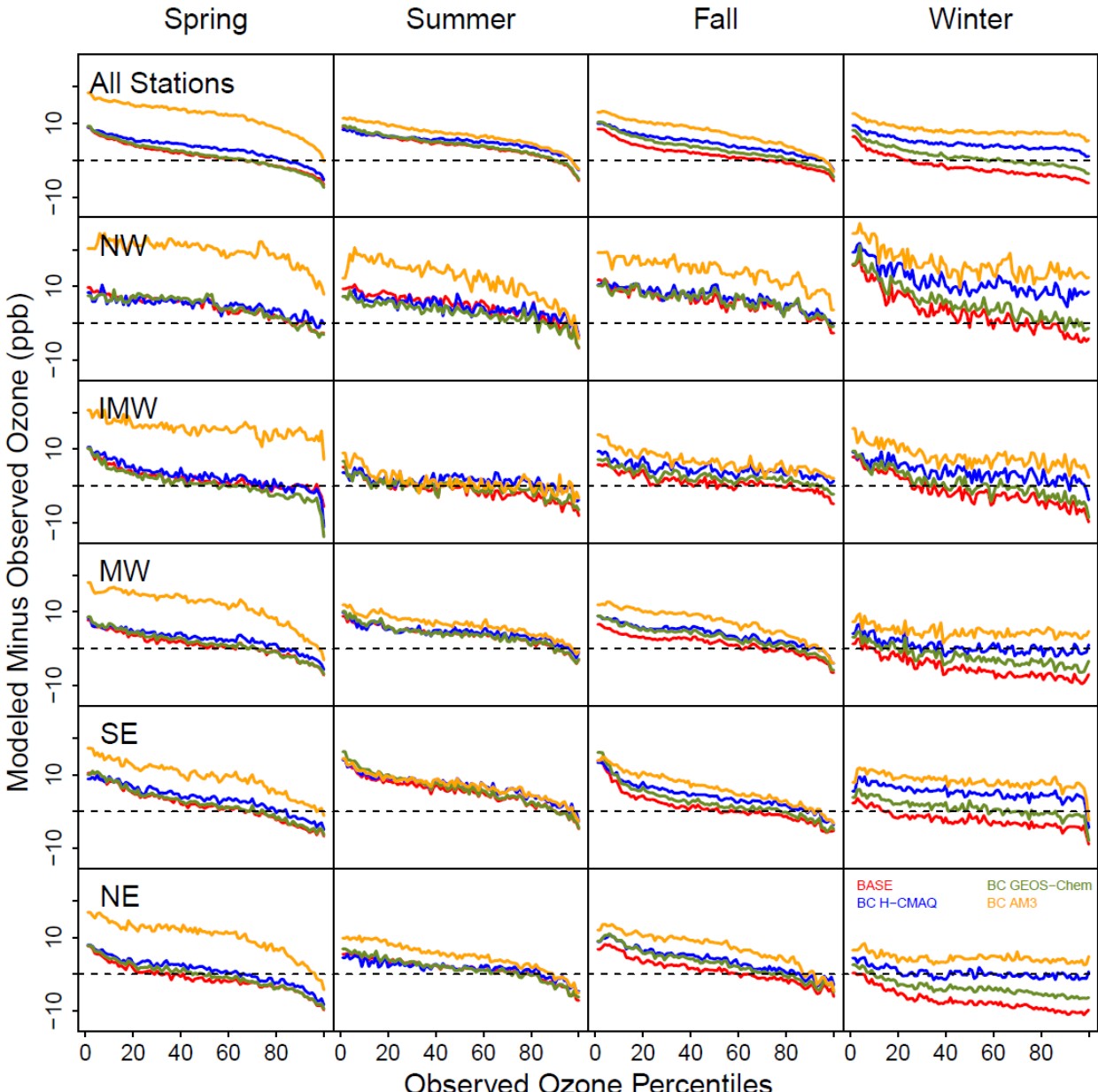

**Figure 13. Paired-in-time differences between observed and modeled MDA8 ozone at AQS stations for each season and analysis region. Model results are for BASE (red), BC H-CMAQ (blue), BC GEOS-Chem (green), and BC AM3 (orange). For each season and region, the observed MDA8 ozone concentrations were rank ordered at each station. Next, differences between CMAQ simulations and observations were computed for each observed percentile by selecting the model value corresponding to the date of the observed percentile. Finally, the median value of these paired-in-time differences across all AQS stations in a given season and region was then computed for each observed percentile and is shown in this figure.**