# Peer review of "Impacts of Different Characterizations of Large-Scale Background on Simulated Regional-Scale Ozone Over the Continental United States"

_Atmospheric Chemistry and Physics, 2017_

## Referee Comment (RC1) · Anonymous Referee #1 · 6 Sep 2017

**Review of Hogrefe et al. for Atmospheric Chemistry and Physics**

*General Comments*

In "Impacts of Different Characterizations of Large-Scale Background on Simulated Regional-Scale Ozone Over the Continental United States", Hogrefe et al. evaluated ozone concentrations at the surface and throughout the column from the Community Multiscale Air Quality (CMAQ) model against observations for 2010 with a careful experimental design that investigated the influence of modeled background ozone concentrations. Additionally, they calculated the contribution to the modeled ozone concentrations of different information used by the model (i.e., boundary conditions, emissions) with a brute force approach as well as processes within the model with an instrumented modeling approach. Leveraging results of the Air Quality Model Evaluation International Initiative (AQMEII) and Task Force on Hemispheric Transport of Air Pollution (TF-HTAP) projects, they quantified the impacts to ozone concentrations at the surface and ozone burdens at different heights aloft of using boundary conditions from four different hemispheric or global models.

The authors showed that the choice of global or hemispheric model for boundary conditions has potential to influence substantially the regulatory metric for ozone on the regional scale on an individual day and the model performance metrics, including the direction of bias. The language and structure of the manuscript are impeccable. The manuscript clearly describes the scope of the investigation, places it in the context of previous research as well as the AQMEII and TF-HTAP efforts, highlights novel results of the analysis, and suggests future research directions this analysis uncovered. I recommend this manuscript for publication in Atmospheric Chemistry and Physics with only editorial changes suggested.

*Specific Comments*

| Line | Comment |
| --- | --- |
| p. 10, l. 14 | "regional-scale simulation". Consider adding "especially near boundaries." The following paragraph highlights the different performance further inland, so it seems important to highlight that this statement is pertinent especially near the boundaries. |
| p. 10, l. 29 | Please change "Figure" to "figure". |
| p. 12, l. 8 | Might "downward" have been intended to be "downwind"? |
| Table 4c | Is the orange in this table a different color than in Tables 4a and 4b? |
| Figure 3 | "right row", "left row" are likely intended to be "right column", "left column". |
| Figures 3, 6, 9, 11, 13 | These figures nicely represent a single statistical metric (e.g., median, mean) for the information shown; however, more information could be conveyed if the standard deviation about the mean or the 5%/95% about the median were displayed with shading or error bars. Could this additional information please be added? |
| Figure 8 | The units seem to have been cut off of the colorbar for these figures. Please include them. Also, for the sake of ease in comparison of the regional figures with the global |

ones, would it be possible to have the y-axes reach the same extent in each but leave white (e.g., a bar across the bottom in the global, a bar across the top in the regional) to indicate that the model did not calculate values at those pressure levels?

---

## Referee Comment (RC2) · Anonymous Referee #2 · 31 Dec 2017

Review of Hogrefe et al., 2017

The authors present results from a number of 12 km WRF-CMAQ base and sensitivity simulations and process analysis related to CONUS ozone in 2010. A great amount of model evaluation (against AQS, CASTNET and sondes observations) work was carried out. The paper does contain a number of highlights and is highly relevant to AQMEII/HTAP2, so I think is publishable after the following comments are addressed.

Specific comments:

[Figure]

1. Table 1 is helpful. For some cases of species mapping, not all CMAQ chemicals had a match from its boundary condition model - Did the authors use a constant (default value)? How was the mapping of aerosol species handled?

I'd like to see some comments on how the completeness of boundary condition model species and the species mapping approach may have contributed to CMAQ model errors (and CMAQ-boundary condition model discrepancies). When we evaluate a given global model's suitability of being used as CMAQ (and other regional CTMs) boundary conditions, should the similarity of CMAQ/boundary condition model's chemistry be considered as an important factor?

2. P5, L7: "Note that these analysis regions do not cover the entire modeling domain." Could the authors explain why they focus on the selected five regions for some of their analyses? The paper does also show the model behavior over other CONUS regions (e.g., Figures 7, 10, 12, and elsewhere). Perhaps an additional panel (named as "entire CONUS" or "all other CONUS regions") could be added to Figures 3 and 13?

3. P5, L7: some clarification is needed on how observations and model data are paired. This is particularly important for understanding the AQS-model evaluation as the numbers of observations in each grid can differ substantially, and can be quite large for some models.

4. Section 2 and Table 2: I suggest adding some brief introductions on non-anthropogenic emissions used in CMAQ and its boundary condition models. These would help us understand each model's performance presented later, and support "the treatment of vertical mixing, lightning emissions, chemistry, deposition and biogenic emissions" in P10, L24-25 and other related statements in the results section.

5. I like the way vertical grid of CMAQ is introduced in P7, L16, i.e., listing the number of vertical layers by altitude range. I suggest adding such information for other models which would nicely complement Figure 1.

6. The authors suggested in P17, L6-8 that "Future work analyzing long-term simulations from multiple global models linked to corresponding regional-scale simulations would be beneficial in better constraining the effects of large-scale interannual variability on regional-scale ozone burdens." I suggest adding some supporting sentences here, e.g., including the differences of ozone and its variability in CMAQ and its boundary condition models (e.g., GEOS-Chem and BC GEOS Chem; AM3 and BC AM3; HCMAQ and BC HCMAQ; C-IFS and BC C-IFS). Such information is included in quite a few figures and discussions but should also be well summarized here (and perhaps in abstract as well if there is space).

Minor issues:

1. In Figure 2 caption or in P5 in text: define lat/lon ranges of each box (region) and numbers of observation sites within each box; provide lat/lon of the ozonesonde sites; Showing the names of the ozonesonde sites in Figure 2 would also be helpful.

2. Define the seasons somewhere.

3. Figure 4, x axis label: is it possible to write it as a math equation?

4. Figures 7, 11: "i" appears to be "l" in the figure captions

5. Unit is missing in Figures 7, 8, 10, 12. Both ppb and ppbV (e.g., Figure 9) are used in the paper and it'd be better to just stick to one of them throughout the paper.

6. Typo in P18, L3: Fiore et al. (2004) Fiore et al. (2014)

7. Typo in Table 4c: BC AM4[56] in the second column should be "BC AM3"

8. Use subscripts for chemical species (e.g., in Table 1)

9. Add data sources of AQS, CASTNET, ozonesondes in the "Acknowledgements and Disclaimer" section

---

## Author Comment (AC1) · 19 Jan 2018

**Response to Comments by Reviewer 1**

General Comments

In "Impacts of Different Characterizations of Large-Scale Background on Simulated Regional-Scale Ozone Over the Continental United States", Hogrefe et al. evaluated ozone concentrations at the surface and throughout the column from the Community Multiscale Air Quality (CMAQ) model against observations for 2010 with a careful experimental design that investigated the influence of modeled background ozone concentrations. Additionally, they calculated the contribution to the modeled ozone concentrations of different information used by the model (i.e., boundary conditions, emissions) with a brute force approach as well as processes within the model with an instrumented modeling approach. Leveraging results of the Air Quality Model Evaluation International Initiative (AQMEII) and Task Force on Hemispheric Transport of Air Pollution (TF-HTAP) projects, they quantified the impacts to ozone concentrations at the surface and ozone burdens at different heights aloft of using boundary conditions from four different hemispheric or global models.

The authors showed that the choice of global or hemispheric model for boundary conditions has potential to influence substantially the regulatory metric for ozone on the regional scale on an individual day and the model performance metrics, including the direction of bias. The language and structure of the manuscript are impeccable. The manuscript clearly describes the scope of the investigation, places it in the context of previous research as well as the AQMEII and TF-HTAP efforts, highlights novel results of the analysis, and suggests future research directions this analysis uncovered. I recommend this manuscript for publication in Atmospheric Chemistry and Physics with only editorial changes suggested.

*Response: We would like to thank the reviewer for the overall positive assessment of our manuscript. We would also like to thank the reviewer for the careful review and helpful suggestions which have led to improvements in some of the figures. Our responses to the specific reviewer comments and the changes incorporated in the revised manuscript are shown below in italics.*

Specific Comments

Comment: p. 10, l. 14 "regional-scale simulation". Consider adding "especially near boundaries." The following paragraph highlights the different performance further inland, so it seems important to highlight that this statement is pertinent especially near the boundaries.

*Response: This suggested change has been incorporated into the revised manuscript.*

Comment: p. 10, l. 29 Please change "Figure" to "figure".

*Response: Thank you for catching this typo, it has been corrected in the revised manuscript.*

Comment: p. 12, l. 8 Might "downward" have been intended to be "downwind"?

*Response: Thank you for catching this typo, it has been corrected in the revised manuscript.*

Comment: Table 4c Is the orange in this table a different color than in Tables 4a and 4b?

*Response: No, the colors are the same. However, upon reviewing Tables 3 and 4b, we noticed that the Normalized Mean Error (NME) values for the BASE simulation included in those tables were incorrect. Specifically, the NME values for the BC H-CMAQ were shown instead of the NME values for BASE in these tables, this has been corrected in the revised manuscript.*

Comment: Figure 3 "right row", "left row" are likely intended to be "right column", "left column".

*Response: Thank you for catching this typo, it has been corrected in the revised manuscript.*

Comment: Figures 3, 6, 9, 11, 13 These figures nicely represent a single statistical metric (e.g., median, mean) for the information shown; however, more information could be conveyed if the standard deviation about the mean or the 5%/95% about the median were displayed with shading or error bars. Could this additional information please be added?

*Response: We appreciate this suggestion and considered it for all the figures listed by the reviewer.*

*For Figure 3 and 13, the variability that could be depicted in addition to the median would represent the spatial variability of model-observation differences across all stations in a given region for a given observed percentile. We prepared versions of these figures that included dashed lines for the 25$^{th}$ and 75$^{th}$ percentile values but decided not include them in the revised manuscript because (i) we found them to be too cluttered especially for the paired-in-time analysis and (ii) we did not consider the additional information on spatial variability contained in these alternate versions to add additional insights for the discussion. However, for completeness, these alternate versions of the figures are included at the end of these responses so that they are available to interested readers as part of the manuscript discussion.*

*For Figure 6, the average represents a temporal average (monthly) of spatially summed hourly ozone column mass values or spatially averaged hourly surface ozone mixing ratios. For the revised version, we updated the figure to include dashed lines that represent the 5$^{th}$ and 95$^{th}$ percentile of the hourly values used to compute the monthly average values.*

*For Figure 9, we considered it too confusing to add more lines or shading since the panels already include four solid and four dashed lines (representing global and regional-scale model results, respectively) as well as a shaded range (depicting the range of observed values averaged over the different altitude ranges represented by the specific model layers at which the individual model values were extracted). Thus, no alternate version of this figure was prepared.*

*For Figure 11, the issue of temporal variability is already addressed by comparing panels c)-d) which show monthly average model-observation differences with panels e)-f) which show the corresponding daily values. The issue of spatial variability of model-observation differences is already addressed in Figure 12 which shows these differences for spring at summer at CASTNET stations. Thus, no alternate version of this figure was prepared.*

Comment: Figure 8 The units seem to have been cut off of the colorbar for these figures. Please include them. Also, for the sake of ease in comparison of the regional figures with the global ones, would it be possible to have the y-axes reach the same extent in each but leave white (e.g., a bar across the bottom in the global, a bar across the top in the regional) to indicate that the model did not calculate values at those pressure levels?

*Response: Thank you for this helpful suggestion, Figure 8a-b has been updated to use the same extent of the vertical axes in all panels. Missing units have been added to the color-bar of Figures 8a-b as well as to the color-bars of Figures 7, 10, 12, and S1. Furthermore, we have also added the following text to Section 3.2.1 when introducing Figure 8a-b*

*"Note that even though observations and large-scale model predictions (except H-CMAQ) are available for higher altitudes (see Figure 1), only values up to the highest model level below 50 mb were extracted for these figures to be comparable to the output from the regional-scale CMAQ simulations (specifically, C-IFS values were only extracted up to layer 38, GEOS-Chem values were only extracted up to layer 37, and AM3 values were only extracted up to layer 26 for this comparison). For easier comparison between models and sites, all figures use a common vertical pressure range of 1025 mb to 50 mb even though this full range is not covered at all sites and by all models."*

[Figure]

*Alternate version of Figure 3 including dashed lines representing the 25th and 75th percentiles of model-observed differences across all stations in a given region for a given observed percentile. The solid lines represent the median across all stations for a given region and given observed percentile.*

[Figure]

*Alternate version of Figure 13 including dashed lines representing the 25$^{th}$ and 75$^{th}$ percentiles of model-observed differences across all stations in a given region for a given observed percentile. The solid lines represent the median across all stations for a given region and given observed percentile.*

---

## Author Comment (AC2) · 19 Jan 2018

**Response to Comments by Reviewer 2**

The authors present results from a number of 12 km WRF-CMAQ base and sensitivity simulations and process analysis related to CONUS ozone in 2010. A great amount of model evaluation (against AQS, CASTNET and sondes observations) work was carried out. The paper does contain a number of highlights and is highly relevant to AQMEII/HTAP2, so I think is publishable after the following comments are addressed.

*Response: We would like to thank the reviewer for the overall positive assessment of our manuscript. We would also like to thank the reviewer for the careful review and helpful comments and suggestions which have led to an improved manuscript. Our responses to the specific reviewer comments and the changes incorporated in the revised manuscript are shown below in italics.*

Specific comments:

Comment: 1. Table 1 is helpful. For some cases of species mapping, not all CMAQ chemicals had a match from its boundary condition model - Did the authors use a constant (default value)? How was the mapping of aerosol species handled? I'd like to see some comments on how the completeness of boundary condition model species and the species mapping approach may have contributed to CMAQ model errors (and CMAQ-boundary condition model discrepancies). When we evaluate a given global model's suitability of being used as CMAQ (and other regional CTMs) boundary conditions, should the similarity of CMAQ/boundary condition model's chemistry be considered as an important factor?

*Response: Yes, when a CMAQ species was not available from a given large-scale model, a constant default value was used for that species. Information on the mapping of aerosol species has been added to the revised manuscript. In terms of whether similarity between the chemistry used in the large-scale model and regional CMAQ should be a consideration when assessing the suitability of a given large-scale model to provide boundary conditions for regional CMAQ, we believe that this question fits in the broader context of our discussion of future research directions in Section 4, specifically our discussion of model consistency across scales. Thus, in response to this comment, we have updated Sections 2 and 4 as follows:*

*"A list of the gas phase species mapped between the large-scale models and CMAQ is shown in Table 1 and a depiction of the vertical layers used in the large-scale models and regional CMAQ simulations is provided in Figure 1. Sulfate, nitrate, ammonium, elemental and organic carbon aerosols were available from all large-scale models while CMAQ trace element aerosol concentrations were estimated from large-scale model dust and sea-salt concentrations except in the case of H-CMAQ which used the same aerosol mechanism as the regional-scale CMAQ simulations. CMAQ species not available from the large scale models were obtained from the time-invariant CMAQ default profile (available at [https://github.com/USEPA/CMAQ/blob/5.0.2/models/BCON/prof_data/cb05_ae6_aq/bc_profile_CB05.dat](https://github.com/USEPA/CMAQ/blob/5.0.2/models/BCON/prof_data/cb05_ae6_aq/bc_profile_CB05.dat), last accessed January 10, 2018)."*

*"The results shown in Section 3 (e.g. Figures 8-9) strongly suggest that differences in the mid-tropospheric ozone mixing ratios simulated by the large-scale models were the main driver of ozone differences between the corresponding regional-scale CMAQ simulations. However, differences in other species such as PAN, differences in the availability of a complete set of CMAQ species from all large-scale*

*models (see Table 1), and inconsistencies in chemical speciation between the large-scale models and regional-scale CMAQ may also have contributed to the ozone differences between the regional-scale CMAQ simulations. Thus, while linking output from available global or hemispheric models to regional-scale models despite such differences represents current best practices in the regional-scale air quality modeling community, additional research should be geared towards developing modeling frameworks that enable a consistent representation of model processes, species, and vertical grid representation across scales. An example of such efforts is the ongoing work to extend CMAQ to hemispheric scales (Mathur et al., 2017). Ensuring such consistency does not in itself guarantee improved model performance but would allow for more targeted diagnostic model evaluation aimed at specific processes which is more challenging when linking together different modeling systems. To achieve such consistency, future work should also be directed toward developing and implementing scale-dependent treatment for atmospheric chemistry in next-generation global dynamic models with variable grid resolution features such as the Model for Prediction Across Scales (MPAS) (Skamarock et al., 2012) and the Finite-Volume Cubed-Sphere Dynamical Core (FV3) model (Harris and Lin, 2013)."*

Comment: 2. P5, L7: "Note that these analysis regions do not cover the entire modeling domain." Could the authors explain why they focus on the selected five regions for some of their analyses? The paper does also show the model behavior over other CONUS regions (e.g., Figures 7, 10, 12, and elsewhere). Perhaps an additional panel (named as "entire CONUS" or "all other CONUS regions") could be added to Figures 3 and 13?

*Response: Thank you for this comment and suggestion. We have added results for the entire domain to Tables 3 – 4 and Figures 3, 13, S3 and S4. We have also added the following information to Section 2 in the revised manuscript:*

"*Model performance evaluation was performed both across the entire domain (1207 AQS monitors and 79 CASTNET monitors) and separately for five sub-regions that are characterized by differences in their proximity to the domain boundaries, elevation, and relative abundance of anthropogenic and biogenic emissions: Northwest (NW) (41 AQS monitors and 2 CASTNET monitors), Intermountain West (IMW) (53 AQS monitors and 7 CASTNET monitors), Midwest (MW) (195 AQS monitors and 13 CASTNET monitors), Southeast (SE) (166 AQS monitors and 13 CASTNET monitors), and Northeast (NE) (204 AQS monitors and 15 CASTNET monitors)."*

Comment: 3. P5, L7: some clarification is needed on how observations and model data are paired. This is particularly important for understanding the AQS-model evaluation as the numbers of observations in each grid can differ substantially, and can be quite large for some models.

*Response: The following information has been added to Section 2 in the revised manuscript:*

"*Furthermore, each monitored value was paired with the corresponding model value based on the model grid cell in which the monitor was located. In particular, multiple observations within the same grid cells were not averaged because the definition of the horizontal grids varied between all the simulations analyzed in this study.*"

Comment: 4. Section 2 and Table 2: I suggest adding some brief introductions on nonanthropogenic emissions used in CMAQ and its boundary condition models. These would help us understand each model's performance presented later, and support "the treatment of vertical mixing, lightning emissions, chemistry, deposition and biogenic emissions" in P10, L24-25 and other related statements in the results section.

Response: Thank you for this suggestion. We have added the following information to Section 2 and also have updated the list of references accordingly:

"However, non-anthropogenic emissions were not harmonized across the global and regional-scale simulations. As described in Flemming et al. (2015), the C-IFS simulations used lightning emissions based on the parameterization introduced in Meijer et al. (2001), biogenic emissions calculated with version 2.1 of the Model of Emissions of Gases and Aerosols from Nature (MEGAN) (Guenther et al., 2006), and biomass burning emissions produced by the Global Fire Assimilation System (GFAS) version 1 (Kaiser et al., 2012). The H-CMAQ simulations used climatological biogenic and lightning emissions from the Global Emission Inventory Activity (GEIA) dataset (Guenther et al., 1995; Price et al., 1997) and biomass burning emissions from version 4.2 of the Emission Database for Global Atmospheric Research (EDGAR) (European Commission, 2011). The GEOS-Chem simulations used lightning emissions based on the methodology described in Murray et al. (2012), biogenic emissions calculated with MEGAN version 2.1, and biomass burning from version 3 of the Global Fire Emissions Database (GFED) (Randerson et al., 2013; van der Werf et al., 2006). The AM3 simulations used lightning emissions based on the parameterization introduced in Horowitz et al. (2003), biogenic emissions calculated with MEGAN version 2.1, and biomass burning emissions from the Fire INventory from NCAR (FINN) (Wiedinmyer et al., 2011). The regional-scale CMAQ simulations did not include lightning emissions, calculated biogenic emissions using version 3.14 of the Biogenic Emission Inventory System (BEIS) (Pierce et al., 1998; Vukovich et al., 2002; Schwede et al., 2005) and used 2010 wildfire emissions as described in Pouliot et al. (2015)."

Comment: 5. I like the way vertical grid of CMAQ is introduced in P7, L16, i.e., listing the number of vertical layers by altitude range. I suggest adding such information for other models which would nicely complement Figure 1.

Response: Thank you for this suggestion. To address the reviewer's comment and to more clearly depict how the vertical structures of the large-scale models relate to the analysis in Figures 5 and 6, we have added layer indices for each model to Figure 1 and have also included horizontal dashed lines at 750mb, 250mb, and 50mb. We believe that including this additional information in the figure enables the reader to easily compare the vertical discretization used by the different models in the pressure ranges of interest. The figure caption has been updated as follows:

"Depiction of the vertical levels used in the four different large-scale models and the regional CMAQ model analyzed in this study. The pressure values were extracted for a location near the southwestern corner of the 12km CMAQ modeling domain and represent annual average values for 2010 at the

*midpoint of each vertical level. The dashed lines delineate the three pressure ranges (surface – 750 mb, 750 mb – 250 mb, and 250 mb – 50 mb) used for vertical integration in subsequent analyses."*

Comment: 6. The authors suggested in P17, L6-8 that "Future work analyzing long-term simulations from multiple global models linked to corresponding regional-scale simulations would be beneficial in better constraining the effects of large-scale inter-annual variability on regional-scale ozone burdens." I suggest adding some supporting sentences here, e.g., including the differences of ozone and its variability in CMAQ and its boundary condition models (e.g., GEOS-Chem and BC GEOS Chem; AM3 and BC AM3; HCMAQ and BC HCMAQ; C-IFS and BC C-IFS). Such information is included in quite a few figures and discussions but should also be well summarized here (and perhaps in abstract as well if there is space).

*Response: This sentence is meant as caveat given that the analysis presented in our manuscript was based on only one year of simulations and as motivation for performing multi-year simulations with both large-scale and regional models to better quantify the effects of inter-annual variability. We do not believe that the results of our current analysis can provide any insights into these effects. In the revised manuscript, we have expanding this sentence in Section 4 as follows to clarify that this should be the focus of future work rather than something that can be obtained from our current work:*

*"Future work analyzing multi-year simulations from multiple global models linked to corresponding regional-scale simulations would be beneficial in better constraining the effects of large-scale inter-annual variability on simulated regional-scale ozone burdens and the inter-annual variability of contributions from large-scale ozone to surface ozone especially during time periods of elevated concentrations"*

Minor issues:

Comment:1. In Figure 2 caption or in P5 in text: define lat/lon ranges of each box (region) and numbers of observation sites within each box; provide lat/lon of the ozonesonde sites; Showing the names of the ozonesonde sites in Figure 2 would also be helpful.

*Response: Figure 2 has been updated to include the names of the ozonesonde sites. Information on the latitude, longitude and elevation of each ozonesonde site and the number of AQS and CASTNET sites in each analysis region has been added to Section 2 in the revised manuscript. Because the rectangular analysis regions are defined on the regional CMAQ modeling grid which uses a Lambert Conformal projection, their bounds are not specified by latitude and longitude ranges. While latitude and longitude values could be provided for each corner of the analysis domains, we believe the graphical depiction of these domains in Figure 1 should be sufficient.*

Comment:2. Define the seasons somewhere.

*Response: The definition has been added to Section 2 in the revised manuscript.*

Comment: 3. Figure 4, x axis label: is it possible to write it as a math equation?

*Response: The axis label has been updated as suggested in the revised manuscript.*

Comment: 4. Figures 7, 11: "i" appears to be "l" in the figure captions

*Response: This impression appears to have been created when pasting the figures into the same document containing the manuscript text and tables since some resolution may have been lost. The original figures which will be provided to the publisher clearly show the correct letter.*

Comment: 5. Unit is missing in Figures 7, 8, 10, 12. Both ppb and ppbV (e.g., Figure 9) are used in the paper and it'd be better to just stick to one of them throughout the paper.

*Response: The figures have been updated to include the units on the colorbar. The use of "ppb" has been made consistent throughout the manuscript and figures.*

Comment: 6. Typo in P18, L3: Fiore et al. (2004) Fiore et al. (2014)

*Response: Thank you for catching this typo, it has been corrected in the revised manuscript.*

Comment: 7. Typo in Table 4c: BC AM4[56] in the second column should be "BC AM3"

*Response: Thank you for catching this typo, it has been corrected in the revised manuscript.*

Comment: 8. Use subscripts for chemical species (e.g., in Table 1)

*Response: This change has been made in the revised manuscript.*

Comment: 9. Add data sources of AQS, CASTNET, ozonesondes in the "Acknowledgements and Disclaimer" section

*Response: URLs with the datasources have been added to the revised manuscript.*